# CausalNovo: Advancing De Novo Peptide Sequencing via a Causality-Informed Framework

## Abstract

*De novo* peptide sequencing is a foundational computational technique in proteomics, which is critical for discovering and characterizing novel peptides and proteins within complex biological systems. To predict peptide sequences directly from tandem mass spectra, mainstream deep learning approaches aim to model the relationship between mass spectra and corresponding peptides. However, these models face significant challenges, particularly under noisy conditions. These deep learning models often capture superficial correlations within noisy spectral data, failing to identify the underlying causal mechanisms that link true signal fragment ions to peptide sequences. Consequently, these models tend to learn spurious associations that cannot generalize in practice, where noise peaks are prone to change due to different co-elutions or chemical contaminants. To tackle this, we introduce **CausalNovo**, a model-agnostic framework designed to learn the causal representations of mass spectra in peptide sequencing models by focusing on signal fragment ions. Specifically, grounded in two practical and general principles, *independence* and *sufficiency*, CausalNovo employs causal interventions and information-theoretic objectives to disentangle causal representations from spurious noise peaks. Extensive experiments on three public datasets show that CausalNovo effectively generalizes across varying Noise Signal Ratios (NSR) and remains relatively stable against non-causal peak changes. Consequently, CausalNovo yields consistent and significant performance gains of up to 10% in amino acid, peptide, and PTM-level performance. Code is available at anonymous link.

## 1 Introduction

Mass spectrometry-based proteomics (Aebersold & Mann, 2003; Strauss et al., 2024; Guo et al., 2025) has revolutionized large-scale protein analysis, providing unprecedented insights into complex biological systems and processes (Lee et al., 2007; Nowinski et al., 2012; Lin et al., 2020; Wenk et al., 2024). A cornerstone of this field, bottom-up proteomics (Zhang et al., 2013; Frejno et al., 2025), focuses on identifying the amino acid sequences of peptides, a task commonly referred to as peptide sequencing. Traditionally, this task relies on database search strategies that match experimental spectra to known protein sequences (Eng et al., 1994; Cox & Mann, 2008; Nesvizhskii, 2010). While effective, these approaches are inherently constrained by their dependence on pre-existing reference databases, limiting the discovery of novel peptides and proteins. To overcome this limitation, recent advances in deep learning have enabled *de novo* peptide sequencing (Tran et al., 2017), which directly predicts peptide sequences from tandem mass spectra that record fragment ion patterns of peptides. This paradigm significantly expands the scope of peptide discovery (Tallorin et al., 2018; Jiang et al., 2024; Bassett et al., 2024).

A wide range of *de novo* sequencing methods (Tran et al., 2017; Yilmaz et al., 2022; Xia et al., 2024; Yang et al., 2024; Zhou et al., 2024; Yilmaz et al., 2024; Eloff et al., 2025; Xia et al., 2025) have been developed, showcasing promising performance. However, these methods are fundamentally limited by their statistical nature: they aim to model dependencies between mass spectra and peptides without accounting for the underlying causal mechanisms. This oversight poses significant challenges in real-world scenarios, where tandem mass spectra often contain a high proportion of noisy or irrelevant peaks (Lubec & Afjehi-Sadat, 2007; Zhou et al., 2024; Yang et al., 2024). Such noise can originate from factors like co-eluting peptides (Alves et al., 2008; Chan et al., 2012) or chemical

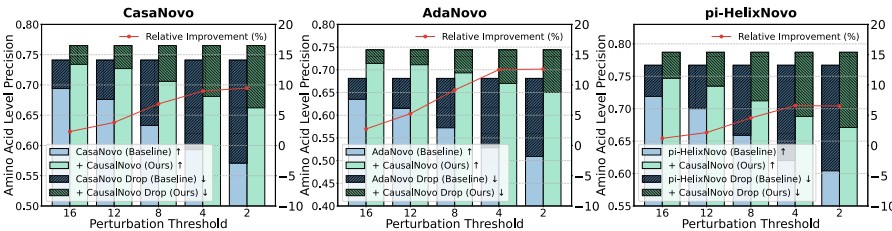

Figure 1: Vulnerability evaluation of baseline models under different perturbation levels. Amino acid precision is reported on the Nine-species (Tran et al., 2017) test set.

contaminants (Krutchinsky & Chait, 2002; Richardson, 2008), introducing spurious correlations that risk misleading statistical models. Critically, these spurious dependencies can obscure the causal relationships between true fragment signals and their corresponding peptides. For example, matrix effects (Trufelli et al., 2011; Panuwet et al., 2016) can lead to consistent co-fragmentation of interfering substances with specific analyte peptides. This results in statistical dependencies that reflect background interactions rather than the intrinsic fragmentation mechanisms of the analyte peptides. Consequently, models trained to exploit these correlations may produce erroneous peptide predictions when the composition of the sample matrix changes due to variations in sample preparation or experimental conditions (Mallick et al., 2007; Pino et al., 2020).

To empirically assess such vulnerability, we conduct a preliminary investigation on the dependency of the existing *de novo* approaches on the non-causal peaks in the mass spectrum. Specifically, for a well-trained sequencing model (*i.e.*, CasaNovo (Yilmaz et al., 2024), AdaNovo (Xia et al., 2024), and $\pi$-HelixNovo (Yang et al., 2024)), we systematically replace noise peaks (*i.e.*, peaks not originating from the analyte peptide) within the tandem mass spectrum to observe performance changes. Following Zhou et al. (2024), signal and noise peaks are identified based on their proximity to the theoretical spectrum (generated by the ground truth peptide sequence to produce b, y and a ions), employing a mass-to-charge ($m/z$) tolerance threshold. As shown in Figure 1, substituting noise peaks leads to a marked decline in amino acid precision, revealing that these models partially depend on spurious correlations. Tightening the $m/z$ tolerance threshold further amplifies this decline, underscoring these models' susceptibility to noise. These results collectively affirm that while current *de novo* sequencing models exhibit encouraging results, their predictive behavior partially relies on spurious correlations with non-causal ions. This motivates the need for a principled framework that incorporates causality into *de novo* peptide sequencing.

Motivated by the analysis above, we propose **CausalNovo**, a model-agnostic framework designed to address the challenges through a principled, causality-informed approach. CausalNovo formalizes the *de novo* peptide sequencing task within the framework of Structural Causal Models (SCMs) (Pearl et al., 2000), which explicitly models the causal mechanisms linking tandem mass spectra to peptide sequence generation. From the SCM, we derive two fundamental principles: *independence*, which ensures the stability of learned representations across varying environments, and *sufficiency*, which guarantees that these representations retain sufficient information for accurate peptide prediction. To operationalize these principles, CausalNovo introduces a Causality Extraction Module (CEM) with multiple information-theoretic objectives to disentangle causal factors from spurious ones in the latent space, promoting causal representations that are both label relevant and invariant to noise and spurious correlations.

We conduct comprehensive experiments to assess the effectiveness of CausalNovo across a wide range of settings. The results show that CausalNovo consistently boosts the performance of all baseline models across amino acid, peptide, and post-translational modification (PTM)-level metrics, achieving impressive improvements of up to 10%. It also surpasses several recent methods by considerable margins. Notably, CausalNovo delivers consistent performance gains under varying Noise Signal Ratios (NSR) across all datasets and baselines. This is attributed to its emphasis on causal signal peaks rather than non-causal information. In summary, these results highlight the potential of CausalNovo to advance the field of MS-based proteomics.

## 2 RELATED WORK

***De Novo* Peptide Sequencing**. Deep learning has significantly advanced *de novo* peptide sequencing. Early methods such as DeepNovo (Tran et al., 2017) and PointNovo (Qiao et al., 2021) introduced deep

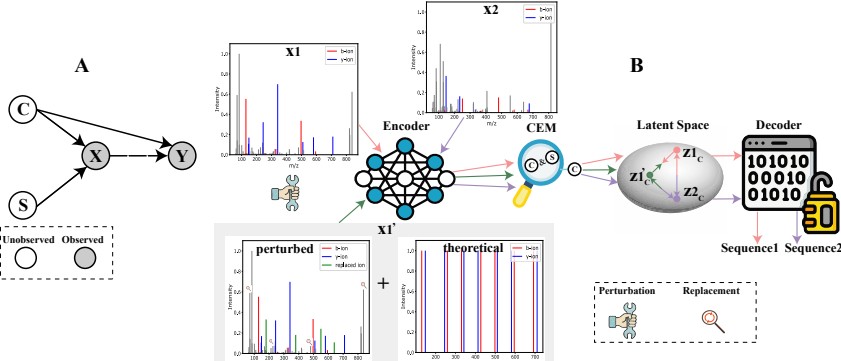

Figure 2: (A) The SCM on *de novo* peptide sequencing; (B) Concept illustration of CausalNovo, which introduces a causality extraction module (CEM) into the baseline sequencing model.

learning neural architectures for sequence prediction, while PepNet (Liu et al., 2023) adopted a fully convolutional neural network design for efficiency. More recently, Transformer-based models have become dominant. For example, CasaNovo (Yilmaz et al., 2022) pioneered the use of Transformer encoder-decoder architectures (Vaswani, 2017), with further improvements from knapsack beam search (Yilmaz et al., 2024) and conditional mutual information training (AdaNovo (Xia et al., 2024)). Other advances include graph modeling (GraphNovo (Mao et al., 2023)), contrastive training between spectra and peptides (ContraNovo (Jin et al., 2024)), spectrum augmentation (π-HelixNovo (Yang et al., 2024)), diffusion-based refinement (InstaNovo (Eloff et al., 2025)), reranking framework (RankNovo (Qiu et al., 2025)), non-autoregressive decoding methods (π-PrimeNovo (Zhang et al., 2025a) and RefineNovo (Zhang et al., 2025b)) and hybrid search strategies (ReNovo (Chen et al., 2025) and SearchNovo (Xia et al., 2025)).

**Causal Machine Learning.** Causal Machine Learning (CausalML) focuses on uncovering true cause-effect relationships in data by modeling the data-generating process with structural causal models (SCMs) (Kaddour et al., 2022; Lv et al., 2022). Unlike traditional machine learning, which focuses primarily on statistical dependencies and risks relying on spurious correlations (Ye et al., 2024; Izmailov et al., 2022), CausalML emphasizes the disentanglement of genuine causal relationships from confounding factors. A major branch of CausalML is causal supervised learning which focuses on learning invariant feature representations guided by causal principles (Kaddour et al., 2022). Recent works have demonstrated that such approachesc can enhance robustness and interpretability (An et al., 2025; Feder et al., 2024; Chen et al., 2022; Sui et al., 2024; Zhao & Zhang, 2024), particularly in biological and medical domains. Given the presence of non-causal fragment ions in spectral data, we extend these principles to *de novo* peptide sequencing, aiming to build models that are both accurate and causally grounded.

## 3 METHODOLOGY

### 3.1 PROBLEM SETUP

The goal of *de novo* peptide sequencing is to infer peptide sequences directly from tandem mass spectra. Formally, each input mass spectrum is represented as $\boldsymbol{x} = \{(m_i, I_i)\}_{i=1}^{n}$, where $m_i$ denotes the mass-to-charge ratio ($m/z$) and $I_i$ the intensity of the $i$-th peak, and $n$ is the number of peaks per spectrum. The $m/z$ values distinguish different fragment ions, such as the prefixes (*i.e.*, $b$-ions) and suffixes (*i.e.*, $y$-ions), derived from the peptide's cleavage. Each spectrum is also associated with a precursor $\boldsymbol{t} = (m_{\text{prec}}, c_{\text{prec}})$, where $m_{\text{prec}} \in \mathbb{R}$ is the precursor mass and $c_{\text{prec}} \in \{1, 2, \ldots, 10\}$ is the charge state. Generally, a *de novo* sequencing model $\rho \circ h$ has an encoder $h : \mathcal{X} \to \mathbb{R}^{n \times d}$ that extracts a representation $\boldsymbol{z}$ for each spectrum to predict the peptide sequence $\hat{\boldsymbol{y}} = \rho(\boldsymbol{z}, \boldsymbol{t})$ with a decoder $\rho : \mathbb{R}^{n \times d} \to \mathcal{Y}$. The ground truth is denoted as $\boldsymbol{y} = \{y_j\}_{j=1}^{L}$, where each $y_j$ belongs to the amino acid vocabulary including 20 canonical amino acids and their PTMs. The peptide length $L$ can vary across different peptides. The goal of peptide sequencing is to train $\rho \circ h$ to sequentially predict each amino acid $y_j$ in the peptide sequence. To this end, auto-aggressive modeling is typically

employed with a cross-entropy loss (Zhou et al., 2024):

$$\mathcal{L}_{\text{CE}}(\boldsymbol{z}, \boldsymbol{t}, \boldsymbol{y}) = -\frac{1}{L} \sum_{j=1}^{L} \log P\left(y_j \mid \boldsymbol{y}_{<j}, \boldsymbol{z}, \boldsymbol{t}\right), \tag{1}$$

where $\boldsymbol{y}_{<j} = \{y_i\}_{i=1}^{j-1}$ represents all amino acids preceding $y_j$ in the peptide sequence.

### 3.2 *De Novo* PEPTIDE SEQUENCING FROM THE CAUSAL VIEW

We formulate the *de novo* peptide sequencing task using a Structural Causal Model (SCM) (Pearl et al., 2000). As illustrated in Figure 2A, the SCM captures the causal relationships among four variables: mass spectrometry data $X$, causal factors $C$, non-causal factors $S$, and peptide sequence label $Y$. Solid arrows represent the causal-effect relationships between variables, while dashed lines denote statistical association.

Existing methods primarily focus on learning statistical associations between the observable variables $X$ (mass spectra) and $Y$ (peptide sequences), often neglecting the underlying causal pathway between $C$ and $Y$. This oversight leads to models that are susceptible to spurious correlations from non-causal factors $S$. To guide the design of more robust and causally informed models, we draw upon Reichenbach's Common Cause Principle (RCCP) (Reichenbach, 1991; Schölkopf et al., 2021), which elucidates the connection between causal relationships and statistical dependence.

**Reichenbach's Common Cause Principle (RCCP)** (Reichenbach, 1991; Schölkopf et al., 2021): If two observables $X$ and $Y$ are statistically dependent, then there exists a variable $C$ that causally influences both and explains all the dependence in the sense of making them independent when conditioned on $C$.

Based on RCCP, we elaborate on the structural equations of the proposed SCM, where noise variables are omitted for simplicity (Chen et al., 2022; Peters et al., 2017):

$$\begin{aligned} X &= f\left(C, S\right), \quad C \perp S, \\ Y &= g\left(C\right), \end{aligned} \tag{2}$$

where $f$ represents the spectrum generation process, and $g$ corresponds to the labeling process. From the equations, we deduce that the **causal factors** $C$ should satisfy two key properties:

  1) *Independence*: $C$ should be independent of non-causal factors $S$. This ensures that the learned representations remain invariant to variations in background noise or interfering substances, preventing the model from relying on spurious correlations introduced by non-causal ions.

  2) *Sufficiency*: $C$ should provide sufficient predictive power for peptide sequencing. This means that once the causal factors are extracted from the spectrum, they should be sufficient for correct prediction without requiring any auxiliary information.

### 3.3 CAUSALNOVO: A CAUSALITY-INSPIRED PEPTIDE SEQUENCING FRAMEWORK

Building on the proposed SCM and the properties of causal factors, we introduce CausalNovo, a framework to extract causal representations and leverage them for accurate and robust peptide prediction. As shown in Figure 2B, CausalNovo integrates a Causality Extraction Module (CEM) into an existing *de novo* sequencing model. Given the high dimensionality and inherent noise in spectral data, direct causal modeling in the raw spectrum space is challenging. Instead, CausalNovo operates in the latent space, where we adopt a representation-level learning strategy. This approach offers a more practical and effective solution for the disentanglement of causal and spurious factors.

**Disentangling causal representation.** We aim to disentangle the causal and non-causal components from the data representation. Specifically, we utilize the CEM $\delta(\cdot)$ to compute importance scores $\boldsymbol{M} = \sigma\left(\delta\left(\boldsymbol{z}\right)\right) \in \mathbb{R}^{n \times 1}$, where $\sigma$ represents the sigmoid function to ensure the importance score $\boldsymbol{M}$ within $[0, 1]$. Each value in $\boldsymbol{M}$ indicates the contribution of the corresponding peak in $\boldsymbol{z}$ to the label prediction. Peaks with higher scores are considered more likely to contain causal information, while those with lower scores are deemed non-causal. Thus, we can obtain the estimated causal representation $\boldsymbol{z_c}$ and non-causal representation $\boldsymbol{z_s}$ as follows:

$$\boldsymbol{z_c} = \boldsymbol{M} \odot \boldsymbol{z}, \; \boldsymbol{z_s} = (1 - \boldsymbol{M}) \odot \boldsymbol{z}, \tag{3}$$

where $\odot$ represents the Hadamard product.

**Independence.** Based on the independence property of the causal factors $C$, $C$ should remain invariant under the intervention on the non-causal factors $S$, i.e., $P(C|S) = P(C|do(S))$, where $do(\cdot)$ denotes an external intervention on the non-causal factors $S$ (Pearl et al., 2016). However, since $S$ is unobservable, it is challenging to directly perturb $S$. Fortunately, the peptide sequence label $Y$ is observable, enabling us to infer which peaks in the spectrum are likely to be noise. By identifying and replacing these noise peaks, we can simulate an intervention on $S$. Thus, the first learning objective in separating causal factors from spectrum is to ensure that the estimated causal representation remains invariant before and after the causal intervention: $\max_{h,\delta} I\left(z_c; z'_c \mid C\right)$, where $z'_c$ is obtained from the embedding $z'$ of perturbed mass spectrometry data.

**Sufficiency.** To ensure that the estimated causal representation captures sufficient causal information for accurate peptide sequence prediction, we directly supervise it using the peptide sequence label $Y$: $\max_{h,\delta,\rho} I\left(z_c; Y\right)$, The decoder $\rho$ is trained to predict the target peptide sequence $Y$ based solely on $z_c$, thereby enforcing that $z_c$ retains all essential predictive information. However, since $z_c$ and $z_s$ may share certain overlapping information about $Y$, optimizing $I(z_c; Y)$ ensures that the inclusion of a partial contribution from non-causal information within $z_c$ does not affect the optimality (Chen et al., 2022). In other words, as long as $z_c$ captures sufficient information necessary for predicting $Y$, the learning objective remains effective even if some spurious, non-causal information leaks into $z_c$. However, it can reduce $I(z_s; Y)$. To address this issue, we introduce an auxiliary objective that maximizes $I(z_s; Y)$ which can indirectly lead to the purification of $z_c$: $\max_{h,\delta,\rho} I\left(z_s; Y\right)$.

### 3.4 PRACTICAL IMPLEMENTATION

#### 3.4.1 CAUSAL INTERVENTION

We then describe the practical implementation of the causal intervention, specifically the $do$ operation in $P(C|S) = P(C|do(S))$. The key objective of this intervention is to perturb the non-causal ions while preserving the causal relationship between signal ions and peptides. Below, we detail the steps to identify non-causal ions and perform the intervention.

**Localizing Non-Causal Ions.** The first step is to identify noise peaks corresponding to the non-causal ions in each spectrum $x$. Given that the training data includes the peptide sequence label, we can compute the theoretical spectrum $x_{\text{theory}} = \left\{(\widetilde{m}_k, \widetilde{I}_k)\right\}_{k=1}^{K}$ for each peptide, where $K$ is the number of peaks. For any peak $(m_i, I_i)$ in $x$, we compare its $m/z$ value $m_i$ with all peaks in the $x_{\text{theory}}$ to determine whether it corresponds to a non-causal ion. The set of non-causal ions is then localized as:

$$x_{\text{non-causal}} = \left\{(m, I) \mid \min_k|m - \widetilde{m}_k| > \gamma, \forall(m, I) \in x\right\}, \tag{4}$$

where $\gamma$ is a tolerance threshold to account for the sensitivity of mass spectrometry, which prevents the misidentification of causal ions as spurious. Notably, this strategy of identifying signal ions is not only a well-established approach in database search (Tyanova et al., 2016) but has also been widely adopted in the design of deep learning models for *de novo* peptide sequencing (Mao et al., 2023; Klaproth-Andrade et al., 2024; Qiao et al., 2021). This highlights the strategy's role as an effective utilization of established domain knowledge in tandem mass spectrometry.

**Replace-based Perturbation.** Once noise peaks is identified, we apply a replacement-based perturbation strategy to simulate causal intervention. Specifically, for each spectrum, a fraction $\alpha$ of peaks in $x_{\text{non-causal}}$ is randomly replaced, yielding $x_{\text{replace}}$. The replacement targets are sampled from all non-causal ions in a training batch. This simple design offers two advantages. Firstly, it ensures the replacement targets are valid, which are real ions with realistic $m/z$ and intensity values. Moreover, the replacement targets introduce new non-causal factors for the spectrum, effectively simulating real-world scenarios where background substances vary.

**Preserving the Causal Relationship.** To safeguard the causal link between $C$ and $Y$ from being perturbed by the intervention, we introduce an additional causality enhancement strategy. Concretely, we further augment the modified spectrum by incorporating all peaks from $x_{\text{theory}}$. This helps preserve causal relationships that may be disrupted by replacement-based interventions. Accordingly, the intervened spectrum is defined as $x_{\text{intervene}} = x_{\text{replace}} \cup x_{\text{theory}}$, and its representation is obtained via $z' = h(x_{\text{intervene}})$.

Table 1: Comparison with state-of-the-art models in amino acid-level and peptide-level performance. † denotes our retrained results, and others are provided by NovoBench. The best results are marked in **bold**, and the second-best results are underlined.

| Method | Year | Amino Acid-Level Performance | | | | | | Peptide-Level Performance | | | | | |
| | | Nine-species | | Seven-species | | HC-PT | | Nine-species | | Seven-species | | HC-PT | |
| | | Prec. | Recall | Prec. | Recall | Prec. | Recall | Prec. | AUC | Prec. | AUC | Prec. | AUC |
|---|---|---|---|---|---|---|---|---|---|---|---|---|---|
| PEAKS (Ma et al., 2003) | 2003 | 0.748 | - | - | - | - | - | 0.428 | - | - | - | - | - |
| DeepNovo (Tran et al., 2017) | 2017 | 0.696 | 0.638 | 0.492 | 0.433 | 0.531 | 0.534 | 0.428 | 0.376 | 0.204 | 0.136 | 0.313 | 0.255 |
| PointNovo (Qiao et al., 2021) | 2021 | 0.740 | 0.671 | 0.196 | 0.169 | 0.623 | 0.622 | 0.480 | 0.436 | 0.022 | 0.007 | 0.419 | 0.373 |
| InstaNovo (Eloff et al., 2025) | 2025 | 0.420 | 0.395 | 0.192 | 0.176 | 0.289 | 0.285 | 0.164 | 0.123 | 0.031 | 0.009 | 0.057 | 0.034 |
| SearchNovo (Xia et al., 2025) | 2025 | 0.748 | 0.746 | 0.489 | 0.488 | 0.652 | **0.658** | 0.550 | 0.489 | 0.259 | 0.174 | 0.447 | 0.413 |
| CasaNovo (Yilmaz et al., 2024) | 2024 | 0.697 | 0.696 | 0.322 | 0.327 | 0.442 | 0.453 | 0.481 | 0.439 | 0.119 | 0.084 | 0.211 | 0.177 |
| †CasaNovo (Yilmaz et al., 2024) | 2024 | 0.741 | 0.740 | 0.357 | 0.366 | 0.525 | 0.530 | 0.529 | 0.493 | 0.159 | 0.119 | 0.324 | 0.290 |
| + CausalNovo (Ours) | - | 0.765 | 0.766 | 0.477 | 0.478 | 0.635 | 0.639 | 0.564 | 0.528 | 0.245 | 0.200 | 0.459 | **0.426** |
| AdaNovo (Xia et al., 2024) | 2024 | 0.698 | 0.709 | 0.379 | 0.385 | 0.442 | 0.451 | 0.505 | 0.469 | 0.174 | 0.135 | 0.212 | 0.178 |
| †AdaNovo (Xia et al., 2024) | 2024 | 0.681 | 0.681 | 0.403 | 0.405 | 0.492 | 0.496 | 0.473 | 0.439 | 0.189 | 0.149 | 0.289 | 0.254 |
| + CausalNovo (Ours) | - | 0.744 | 0.746 | 0.453 | 0.453 | 0.634 | 0.637 | 0.542 | 0.507 | 0.233 | 0.192 | 0.453 | 0.420 |
| π-HelixNovo (Yang et al., 2024) | 2024 | 0.765 | 0.758 | 0.481 | 0.472 | 0.588 | 0.582 | 0.517 | 0.453 | 0.234 | 0.173 | 0.356 | 0.318 |
| †π-HelixNovo (Yang et al., 2024) | 2024 | 0.765 | 0.752 | 0.465 | 0.462 | 0.532 | 0.537 | 0.509 | 0.431 | 0.218 | 0.156 | 0.301 | 0.261 |
| + CausalNovo (Ours) | - | **0.787** | **0.784** | **0.536** | **0.534** | **0.656** | **0.658** | 0.543 | 0.483 | **0.282** | **0.229** | 0.450 | 0.415 |

Table 2: Comparison with state-of-the-art models in PTM-level performance. † denotes our retrained results, and others are given by NovoBench. The best is marked in **bold**.

| Method | Nine-species | | Seven-species | | HC-PT | |
| | Prec. | Recall | Prec. | Recall | Prec. | Recall |
|---|---|---|---|---|---|---|
| DeepNovo | 0.576 | 0.529 | 0.391 | 0.373 | 0.626 | 0.615 |
| PointNovo | 0.629 | 0.546 | 0.117 | 0.094 | 0.676 | 0.740 |
| InstaNovo | 0.443 | 0.294 | 0.126 | 0.115 | 0.350 | 0.261 |
| SearchNovo | 0.764 | 0.599 | 0.472 | **0.447** | 0.715 | **0.772** |
| CasaNovo | 0.706 | 0.566 | 0.360 | 0.251 | 0.501 | 0.460 |
| †CasaNovo | 0.755 | 0.601 | 0.368 | 0.292 | 0.550 | 0.582 |
| + CausalNovo | **0.791** | 0.608 | 0.503 | 0.422 | 0.671 | 0.741 |
| AdaNovo | 0.652 | 0.570 | 0.448 | 0.321 | 0.552 | 0.482 |
| †AdaNovo | 0.678 | 0.552 | 0.430 | 0.356 | 0.562 | 0.532 |
| + CausalNovo | 0.769 | 0.607 | 0.469 | 0.398 | 0.652 | 0.743 |
| π-HelixNovo | 0.680 | 0.598 | 0.473 | 0.366 | 0.568 | 0.667 |
| †π-HelixNovo | 0.723 | 0.593 | 0.362 | 0.370 | 0.632 | 0.566 |
| + CausalNovo | 0.731 | **0.616** | **0.513** | 0.427 | **0.737** | 0.746 |

Table 3: Cross-species validation on the Nine-species dataset. † denotes our retrained results.

| Species | Method | Amino Acid | | Peptide | |
| | | Prec. | Recall | Prec. | AUC |
|---|---|---|---|---|---|
| Bacillus | †CasaNovo | 0.743 | 0.745 | 0.559 | 0.523 |
| | + CausalNovo | **0.773** | **0.774** | **0.584** | **0.551** |
| Clambacteria | †CasaNovo | 0.754 | 0.752 | 0.544 | 0.509 |
| | + CausalNovo | **0.772** | **0.773** | **0.578** | **0.544** |
| Honeybee | †CasaNovo | 0.761 | 0.757 | 0.554 | 0.519 |
| | + CausalNovo | **0.772** | **0.774** | **0.578** | **0.545** |
| Human | †CasaNovo | 0.769 | 0.770 | 0.575 | 0.540 |
| | + CausalNovo | **0.780** | **0.779** | **0.587** | **0.553** |
| M.mazei | †CasaNovo | 0.754 | 0.754 | 0.558 | 0.532 |
| | + CausalNovo | **0.773** | **0.776** | **0.580** | **0.544** |
| Mouse | †CasaNovo | 0.753 | 0.752 | 0.558 | 0.521 |
| | + CausalNovo | **0.784** | **0.782** | **0.595** | **0.560** |
| Ricebean | †CasaNovo | 0.753 | 0.755 | 0.549 | 0.510 |
| | + CausalNovo | **0.763** | **0.763** | **0.557** | **0.521** |
| Tomato | †CasaNovo | 0.717 | 0.716 | 0.506 | 0.464 |
| | + CausalNovo | **0.748** | **0.752** | **0.545** | **0.506** |

### 3.4.2 OPTIMIZING OBJECTIVE

**Optimizing the Independence.** In practice, $C$ is unobserved, but $Y$ can serve as a proxy for $C$ in Eq. (3.3) due to the stable causal relationship between them (Chen et al., 2022). Moreover, obtaining an exact estimate of Eq. (3.3) can be computationally expensive (Belghazi et al., 2018; Chen et al., 2022). However, contrastive learning offers a practical approach for approximation (Oord et al., 2018; Khosla et al., 2020; You et al., 2020; Chen et al., 2022):

$$I(\boldsymbol{z}_c; \boldsymbol{z}_c' \mid Y) \approx \log \frac{\exp(\text{sim}(\boldsymbol{z}_c, \boldsymbol{z}_c')/\tau)}{\exp(\text{sim}(\boldsymbol{z}_c, \boldsymbol{z}_c')/\tau) + \sum_{\boldsymbol{z}_c'^- \in \mathcal{N}} \exp(\text{sim}(\boldsymbol{z}_c, \boldsymbol{z}_c'^-)/\tau)}, \quad (5)$$

where sim denotes the cosine similarity between aggregated causal representations using mean pooling, $\tau$ is the temperature parameter, $\boldsymbol{z}_c'^-$ denotes the causal representation of the negative sample set $\mathcal{N}$. In our implementation, we simply use the current training batch (excluding $\boldsymbol{z}_c'$) as $\mathcal{N}$. Additionally, inspired by existing contrastive learning studies (Radford et al., 2021), we also incorporate a symmetric strategy to more effectively utilize the training samples. Specifically, the causal representation of the replace-intervened example $\boldsymbol{x}_{\text{replace}}$ is alternately used as the anchor, while the enhanced original example serves as the positive sample for contrastive learning.

**Optimizing Sufficiency and Purification.** Maximizing mutual information is equivalent to minimizing cross-entropy loss (Boudiaf et al., 2020). Thus, the objectives in Eq. (3.3) and Eq. (3.3) can be optimized using amino acid-level cross-entropy loss:

$$I(\boldsymbol{z}_c; Y) \propto -\mathcal{L}_{\text{CE}}(\boldsymbol{z}_c, \boldsymbol{t}, \boldsymbol{y}), \quad I(\boldsymbol{z}_s; Y) \propto -\mathcal{L}_{\text{CE}}(\boldsymbol{z}_s, \boldsymbol{t}, \boldsymbol{y}). \quad (6)$$

Table 4: Ablation study on the components of CausalNovo. CasaNovo is employed as the baseline.

| Baseline | Independence | Purification | Symmetric | Amino Acid-Level | | Peptide-Level | | PTM-Level | |
|---|---|---|---|---|---|---|---|---|---|
| | | | | Prec. | Recall | Prec. | AUC | Prec. | Recall |
| ✓ | ✗ | ✗ | ✗ | 0.741 | 0.740 | 0.529 | 0.493 | 0.755 | 0.601 |
| ✓ | ✓ | ✗ | ✗ | 0.753 | 0.753 | 0.552 | 0.513 | 0.772 | 0.602 |
| ✓ | ✓ | ✓ | ✗ | 0.761 | 0.761 | 0.560 | 0.522 | **0.792** | **0.618** |
| ✓ | ✓ | ✓ | ✓ | **0.765** | **0.766** | **0.564** | **0.528** | 0.791 | 0.608 |

Table 5: Ablation study on the causal intervention.

| Baseline | Replace | Enhance | Drop | Amino Acid-Level | | Peptide-Level | | PTM-Level | |
|---|---|---|---|---|---|---|---|---|---|
| | | | | Prec. | Recall | Prec. | AUC | Prec. | Recall |
| ✓ | ✗ | ✗ | ✗ | 0.741 | 0.740 | 0.529 | 0.493 | 0.755 | 0.601 |
| ✓ | ✓ | ✗ | ✗ | 0.747 | 0.750 | 0.546 | 0.510 | 0.761 | 0.596 |
| ✓ | ✓ | ✓ | ✗ | **0.753** | 0.753 | **0.552** | **0.513** | **0.772** | 0.602 |
| ✓ | ✓ | ✓ | ✓ | **0.753** | **0.754** | 0.551 | 0.511 | 0.765 | **0.605** |

## 4 EXPERIMENT

### 4.1 EXPERIMENTAL SETUP

**Datasets.** Following NovoBench (Zhou et al., 2024), we conduct experiments on Nine-species (Tran et al., 2017), Seven-species (Tran et al., 2017), and HC-PT (Eloff et al., 2025) datasets. For Nine-species and Seven-species, the *yeast* species is used as the test set, while the remaining species are utilized for training and validation purposes.

**Baselines & Metrics.** We integrate our CausalNovo framework with three established sequencing competitors, including CasaNovo (Yilmaz et al., 2024), AdaNovo (Xia et al., 2024), and $\pi$-HelixNovo (Yang et al., 2024). We retrain the baselines with the same configurations. Other competitors included DeepNovo (Tran et al., 2017), PointNovo (Qiao et al., 2021), InstaNovo (Eloff et al., 2025) and SearchNovo (Xia et al., 2025). We evaluate the performance using amino acid-level precision and recall, the peptide-level precision and AUC, and PTM-level precision and recall.

### 4.2 IMPLEMENTATION DETAILS

**Architecture.** The spectrum encoder is a Transformer Encoder with 9 layers. The peptide decoder is a Transformer Decoder with 9 layers. The causal extraction module contains 3 Transformer layers followed by an MLP head. The hidden dimension of all transformer blocks is set to 512, with 8 attention heads and an FFN layer dimension of 1024.

**Training and Inference.** Models are trained with a batch size of 32 for 30 epochs. The learning rate is 5e-4, with a weight decay of 1e-5. The learning rate is linearly increased from zero to the peak value in 100k warm-up steps, followed by a cosine-shaped decay. The maximum peptide length is set to 100, while the minimum length is set to 6. The maximum number of peaks is set to 150, with the minimum $m/z$ value set to 50.52564895 and the maximum $m/z$ value set to 2500. Peaks with normalized intensity values lower than 0.01 are filtered out. The temperature $\tau$ is set to 0.1. During inference, a beam search strategy with a beam size of 5 is utilized for all models. Experiments were performed using an NVIDIA GeForce RTX 4090 GPU.

### 4.3 COMPARISON WITH THE STATE-OF-THE-ARTS

**Amino Acid-Level Comparison.** CausalNovo boosts all baseline models by large margins. Specifically, on the Nine-species dataset, CausalNovo boosts CasaNovo, AdaNovo and $\pi$-HelixNovo by +2.4%, +6.3% and +2.2% in precision. On Seven-species, CausalNovo improves the three baselines by +12.0%, +5.0% and +9.1% in precision, respectively. Additionally, on HC-PT, CausalNovo brings an improvement of +9.0%, +14.2% and +12.4% in precision, respectively.

**Peptide-Level Comparison.** Peptide-level performance is essential for evaluating the practical utility of models, as the main goal of the peptide sequencing task is to accurately predict a complete peptide sequence for each spectrum. Here we show that CausalNovo also improves the peptide precision and AUC on all baselines by large margins. Concretely, CausalNovo improves CasaNovo's precision by +3.5% on Nine-species, which also outperforms SearchNovo by +1.4%. On the other two datasets, $\pi$-HelixNovo + CausalNovo and AdaNovo + CausalNovo achieve the best performance respectively, which also outperforms SearchNovo by considerable margins. We also provide peptide

Figure 3: Vulnerability evaluation on the HC-PT dataset.

Table 6: Analysis of peak distinguish strategies. A more comprehensive set of 18 types of ions is considered. RI means the relative improvement.

| Threshold | Method | Nine-species | | | | HC-PT | | | |
| | | AA.Prec | RI ↑ | Pep.Prec | RI ↑ | AA.Prec | RI ↑ | Pep.Prec | RI ↑ |
|---|---|---|---|---|---|---|---|---|---|
| 8 | CasaNovo | 0.711 | - | 0.501 | - | 0.504 | - | 0.298 | - |
| | + CausalNovo | **0.744** | **1.3%** | **0.542** | **1.2%** | **0.627** | **2.1%** | **0.442** | **4.3%** |
| 4 | CasaNovo | 0.682 | - | 0.470 | - | 0.474 | - | 0.259 | - |
| | + CausalNovo | **0.731** | **3.5%** | **0.525** | **4.1%** | **0.617** | **6.3%** | **0.423** | **12.2%** |
| 2 | CasaNovo | 0.629 | - | 0.420 | - | 0.424 | - | 0.198 | - |
| | + CausalNovo | **0.709** | **7.8%** | **0.501** | **9.3%** | **0.596** | **12.5%** | **0.396** | **25.2%** |
| 1 | CasaNovo | 0.605 | - | 0.397 | - | 0.386 | - | 0.156 | - |
| | + CausalNovo | **0.689** | **8.4%** | **0.482** | **10.3%** | **0.561** | **14.3%** | **0.352** | **28.5%** |

precision-coverage curve comparison in the Appendix ( Figure 6), which shows CausalNovo leads to consistent improvements across coverage levels and datasets.

**PTM-Level Comparison.** Identifying PTMs poses a significant challenge for existing models (Xia et al., 2024). CausalNovo improves the PTM-level performance of all baseline models across all datasets. Notably, CausalNovo enhances the PTM precision of $\pi$-HelixNovo by +15.1% on the Seven-species dataset, which also surpasses the recent SearchNovo by +4.1%.

**Cross-Species Validation.** To evaluate the generalizability, we further conduct leave-one-out cross-species validation experiments, where each species is alternately selected as the test set while the model was trained on the remaining species. Table 3 shows that, in the Nine-species dataset, the CausalNovo framework significantly enhances the CasaNovo baseline across all species. On average, CausalNovo improves peptide precision by +2.6% compared to the baseline model across all species. We also provide cross-species validation on the Seven-species dataset, which show that CausalNovo improves peptide precision by +6.7% (Appendix Table 8).

## 4.4 MODEL ANALYSIS

**Component Analysis.** We evaluate the contribution of each component in CausalNovo through ablation studies on the Nine-species dataset using CasaNovo as the baseline (Table 4). Notably, the sufficiency principle, which corresponds to the amino acid-wise cross-entropy loss, is already included in the baseline model. Incorporating the independence principle enhances the baseline by +1.2%, +1.3%, +2.3%, and +2.0% in amino acid and peptide-level performance. Adding the purification training objective further improves amino acid precision and recall by +0.8% and +0.7%, respectively. Finally, the symmetric training scheme provides an additional +0.4% and +0.5% improvement. This ablation study highlights the effectiveness of each component.

**Ablating the Causal Intervention.** To analyze the contribution of each component in the causal intervention, we then conducted an ablation study on the Nine-species dataset, with results shown in Table 5. The results indicate that the replace-based perturbation strategy improves the CasaNovo baseline by +0.6% in amino acid-level precision and +1.0% in recall. Incorporating the causality enhancement further increases these metrics by +0.6% and +0.3%, respectively. We also tested a random drop operation, where 20% of noise peaks were randomly removed. This operation did not lead to performance improvement. Thus, we used the replace and enhance for the causal intervention.

**Vulnerability Analysis.** Figure 1 and Figure 3 show that the performance of the three baseline models consistently degrades when noise peaks are perturbed in the Nine-species and HC-PT datasets, highlighting their reliance on non-causal ions. In contrast, after incorporating our CausalNovo framework, the performance degradation across all datasets was significantly mitigated under various perturbation thresholds. Specifically, CausalNovo leads to consistent and substantial performance gains ranging from 2.8% to 14.2% across all settings. We further compute the Relative Improvement (RI), defined as the relative performance reduction of CausalNovo compared to the baseline models. As shown in

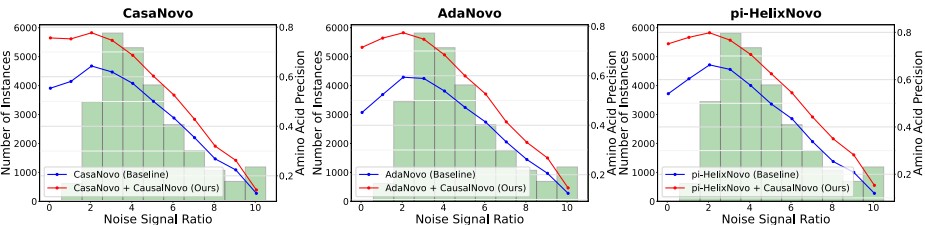

Figure 4: Generalization across varying Noise Signal Ratios (NSR) on the HC-PT dataset.

Figure 3, CausalNovo achieves an average RI of +14.9%, +15.7%, and +13.5% compared to three baseline models on the HC-PT dataset, respectively. These results demonstrate that CausalNovo exhibits a stronger reliance on causal signal peaks instead of non-causal peaks. More results on other datasets can be found in Appendix A.2.

**Analysis of Peak Distinguish Strategies.** Our framework uses three ion types (b, y, a) to distinguish causal from non-causal peaks, as these ions are the most common in benchmark datasets and provide sufficient causal information for sequencing. However, different mass spectra might include diverse ions, potentially affecting model performance. To test this, we further consider a comprehensive set of 18 ion types (including singly and doubly charged ions of $b^+$, $y^+$, $a^+$, $b^{2+}$, $y^{2+}$, $a^{2+}$, and neutral loss variants: -NH3 and -H2O for each ion type), following Mao et al. (2023). This ensures a broader coverage and more accurate non-causal peak estimation. Based on this, we re-evaluate the vulnerability of our model (without retraining) compared to the baseline. As shown in Table 6, replacing non-causal peaks still significantly degrades baseline performance, while CausalNovo achieves a 28.5% relative improvement under the same setup (threshold = 1). Moreover, retraining our framework with all 18 ion types yields similar performance (Appendix Table 12). This is because the original three types of ions already cover most signal ones. These results collectively demonstrate that our framework is both effective and robust across different peak-distinguishing strategies.

**Generalization across Varying Noise Signal Ratios.** We further evaluate the generalizability of CausalNovo across varying noise-signal ratios (NSR), which is defined as the ratio of noise peaks to signal peaks in the mass spectrum (Zhou et al., 2024). The results are shown in Figure 4. By focusing more on causal peaks, CausalNovo effectively mitigates the impact of noise fragment ions on all three baseline models. Specifically, CausalNovo improves the amino acid precision by an average of +10.2%, +13.2%, and +12.0% across different NSRs for the three baseline models, respectively. These results further validates the efficacy of our CausalNovo framework. More results on other datasets and evaluation metrics can be found in Appendix A.2.

**Analysis of Model Attention to Causal Peaks.** We analyze how CausalNovo attends to causal versus non-causal peaks by examining its attention matrices (*i.e.*, interpretable matrices defined by $\pi$-xNovo (Wang et al., 2024)). For each amino acid prediction, we count the number of causal peaks among the model's top three most attended peaks. As shown

Table 7: Analysis of model attention to causal peaks.

| # Causal Peaks | 0 | 1 | 2 | 3 |
|---|---|---|---|---|
| Baseline | 44,585 | 112,463 | 125,773 | 67,453 |
| Ratio | 12.73% | 32.11% | 35.91% | 19.26% |
| + CausalNovo | 37,414 | 78,170 | 117,804 | 114,276 |
| Ratio | 10.76% | 22.48% | 33.88% | 32.87% |

in Table 7, the baseline model completely ignores causal peaks in 12.73% of predictions, while CausalNovo reduces this to 10.76%. Furthermore, only 19.26% of predictions in the baseline fully attend to causal peaks, compared to 32.87% in CausalNovo. We also analyze cases (Appendix Table 14) where the baseline produces incorrect predictions but CausalNovo makes corrections. In these cases, the baseline fails to attend to causal peaks in 14.18% of predictions, whereas CausalNovo reduces this to 5.44%. These results demonstrate that CausalNovo significantly improves attention to causal peaks, leading to substantial performance gains.

## 5 CONCLUSION AND DISCUSSION

This work presents **CausalNovo**, a causality-inspired model-agnostic framework for *de novo* peptide sequencing that encourages learning from true fragment ion signals while mitigating reliance on spurious spectral noise. Extensive experiments across three benchmarking datasets and multiple state-of-the-art baselines demonstrate consistent improvements at the amino acid, peptide, and PTM levels. While CausalNovo introduces negligible inference overhead (less than 1%), it increases approximately 2.3x training time due to the need for multiple forward passes per batch, highlighting

a key limitation. Moreover, our evaluation follows the NovoBench setting, whereas recent methods (e.g., ContraNovo, RankNovo) adopt a more realistic protocol that trains on large-scale external corpora and evaluates on out-of-distribution test sets. Assessing CausalNovo under this protocol would better reflect real-world utility and is a priority for future work. In summary, as a general and extensible framework, CausalNovo lays the groundwork for integrating causality into proteomics and opens promising directions for robust, interpretable modeling in noisy biological settings.

## Ethics Statement

This work adheres to the ICLR Code of Ethics. In this study, no human subjects or animal experimentation was involved. All datasets used, including Nine-species, Seven-species and HC-PT, were sourced in compliance with relevant usage guidelines, ensuring no violation of privacy. We have taken care to avoid any biases or discriminatory outcomes in our research process. No personally identifiable information was used, and no experiments were conducted that could raise privacy or security concerns. We are committed to maintaining transparency and integrity throughout the research process.

## Reproducibility Statement

We have made every effort to ensure that the results presented in this paper are reproducible. All code has been made publicly available in an anonymous repository to facilitate replication and verification. The experimental setup, including training steps, model configurations, and hardware details, is described in detail in the paper. We have also provided a full description of CausalNovo, to assist others in reproducing our experiments. Additionally, all the datasets used in the paper, such as Nine-species, Seven-species, and HC-PT, are publicly available, ensuring consistent and reproducible evaluation results. We believe these measures will enable other researchers to reproduce our work and further advance the field.

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

# A APPENDIX

The appendix is structured as follows: Appendix A.1 introduces the research background of *de novo* peptide sequencing. Appendix A.2 presents additional experimental results along with an in-depth analysis. Finally, Appendix A.3 provides detailed information about the datasets used in this study.

## A.1 RESEARCH BACKGROUND

Proteomics encompasses the comprehensive examination of proteins' structure, function, and interactions within biological systems, with the goal of unraveling cellular processes and biological mechanisms (Blackstock & Weir, 1999). A pivotal aspect of proteomics involves identifying proteins within biological samples. Tandem mass spectrometry has emerged as the principal high-throughput method for protein identification.

As shown in Figure 5, the conventional shotgun proteomics workflow (Wolters et al., 2001) initiates with enzymatic protein digestion, yielding mixture of peptides. These peptides, termed precursors, are then subjected to analysis by a mass spectrometer, which captures their mass-to-charge ($m/z$) ratios in the initial scan (MS1). Subsequently, the peptides undergo fragmentation via methods like collision-induced dissociation (CID)(Yates et al., 1995) and higher-energy collisional dissociation (HCD)(Olsen & Mann, 2004), generating fragments analyzed in a subsequent scan (MS2). During this fragmentation process, peptides cleave randomly along their backbone, giving rise to ions representing the peptide's prefixes (referred to as $b$-ions) and suffixes (known as $y$-ions).

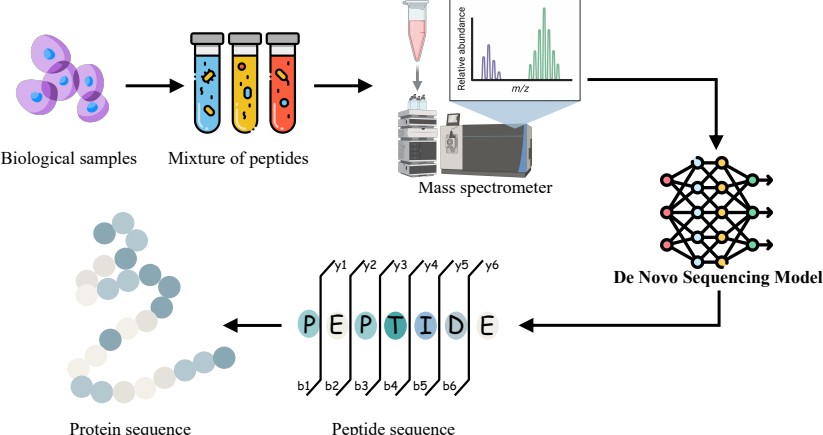

Figure 5: The workflow of shotgun proteomics (Wolters et al., 2001).

The resulting mass spectrum comprises peaks characterized by $m/z$ values and relative abundances of the fragment ions. While $m/z$ values are measured with precision, abundance values, while less accurate, are proportional to the contributing ions' quantity in each peak. *De novo* peptide sequencing involves constructing deep learning models that leverage mass spectrum data and associated precursor information to predict the peptide sequence responsible for the experimental mass spectrum. This prediction is followed by piecing together various peptide segments to ascertain the complete protein sequence.

This entire process is crucial for deciphering the intricacies of the proteome, providing profound insights into the molecular foundations of biological systems and advancing the comprehension of cellular functions and disease mechanisms.

## A.2 MORE RESULTS AND ANALYSIS

**Comparison of Peptide Precision-Coverage Curves.** In Figure 6, we analyze the peptide precision-coverage curves across three datasets. These curves illustrate the trade-off between precision and coverage in different scenarios, providing more insights into model performance. We take CasaNovo

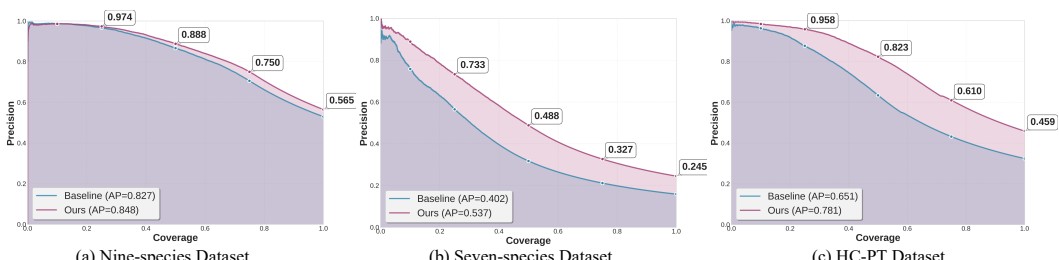

Figure 6: Comparison of peptide precision-coverage curves with CasaNovo. AP represents the average precision.

Table 8: Cross-species validation on the Seven-species dataset. † denotes our retrained results.

| Species | Method | Amino Acid | | Peptide | |
| | | Prec. | Recall | Prec. | AUC |
|---|---|---|---|---|---|
| Celegans | †CasaNovo (Yilmaz et al., 2024) | 0.436 | 0.439 | 0.220 | 0.174 |
| | **+ CausalNovo** | **0.512** | **0.513** | **0.284** | **0.234** |
| Ecoli. | †CasaNovo (Yilmaz et al., 2024) | 0.407 | 0.412 | 0.208 | 0.165 |
| | **+ CausalNovo** | **0.511** | **0.514** | **0.285** | **0.241** |
| Fruitfly | †CasaNovo (Yilmaz et al., 2024) | 0.435 | 0.438 | 0.222 | 0.179 |
| | **+ CausalNovo** | **0.503** | **0.513** | **0.293** | **0.250** |
| Human | †CasaNovo (Yilmaz et al., 2024) | 0.418 | 0.421 | 0.210 | 0.167 |
| | **+ CausalNovo** | **0.507** | **0.509** | **0.292** | **0.247** |
| Mouse | †CasaNovo (Yilmaz et al., 2024) | 0.496 | 0.496 | 0.268 | 0.223 |
| | **+ CausalNovo** | **0.534** | **0.536** | **0.299** | **0.256** |
| Pseudomonas | †CasaNovo (Yilmaz et al., 2024) | 0.425 | 0.437 | 0.227 | 0.188 |
| | **+ CausalNovo** | **0.502** | **0.502** | **0.281** | **0.242** |

as the baseline. The results highlight the effectiveness of our approach across diverse datasets, demonstrating superior precision at varying levels of coverage.

**Cross-species Validation on Seven-species Dataset.** In the Seven-species dataset, the proposed CausalNovo framework also demonstrates impressive generalization ability across species. Specifically, on Ecoli., CausalNovo boosts the baseline model by +10.4%, +10.2%, +7.7%, and +7.6% in four metrics. On average, CausalNovo improves peptide precision by +6.7% compared to the baseline model across all seven species.

**Vulnerability Evaluation on More Datasets** We present the vulnerability evaluation of the three baseline models on the Seven-species dataset, as shown in Figure 7. While the baseline models consistently degrade under noise peak perturbations, our method significantly enhances their performance, demonstrating a substantial improvement.

**Performance across Noise Signal Ratios on More Datasets.** We further present experimental results across varying Noise Signal Ratios (NSR) on three datasets, as shown in Figure 8, Figure 10, and Figure 11. To provide a comprehensive evaluation, we report amino acid-level precision and recall, as well as peptide-level precision. As these results demonstrate, our CausalNovo framework consistently outperforms baseline models across almost all NSR ratios on these datasets. This further highlights the superiority of our CausalNovo framework.

**Comparison with SSL methods.** We also make comparisons of our CausalNovo framework with other self-supervised learning (SSL) frameworks, including the commonly used contrastive learning (Jin et al., 2024), SimSaim (Chen & He, 2021), and MAE (He et al., 2022). As shown in Table 9, simply integrating SSL methods into de novo peptide sequencing does not lead to substantial performance improvement. In contrast, incorporating our proposed causal learning mechanism yields significant performance gains. This demonstrates that the main improvement originates from the causal learning framework rather than the SSL learning signals.

**Configuration of the CEM module.** For the CEM module architecture, we test three architectures, including a simple linear layer, a single transformer encoder layer, and a 3-transformer encoder layer configuration. Our experiments in Table 10 show that our setup performs the best.

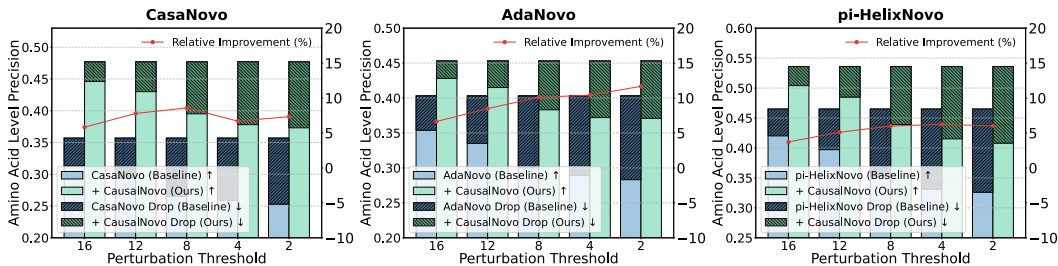

Figure 7: Vulnerability evaluation on the Seven-species dataset.

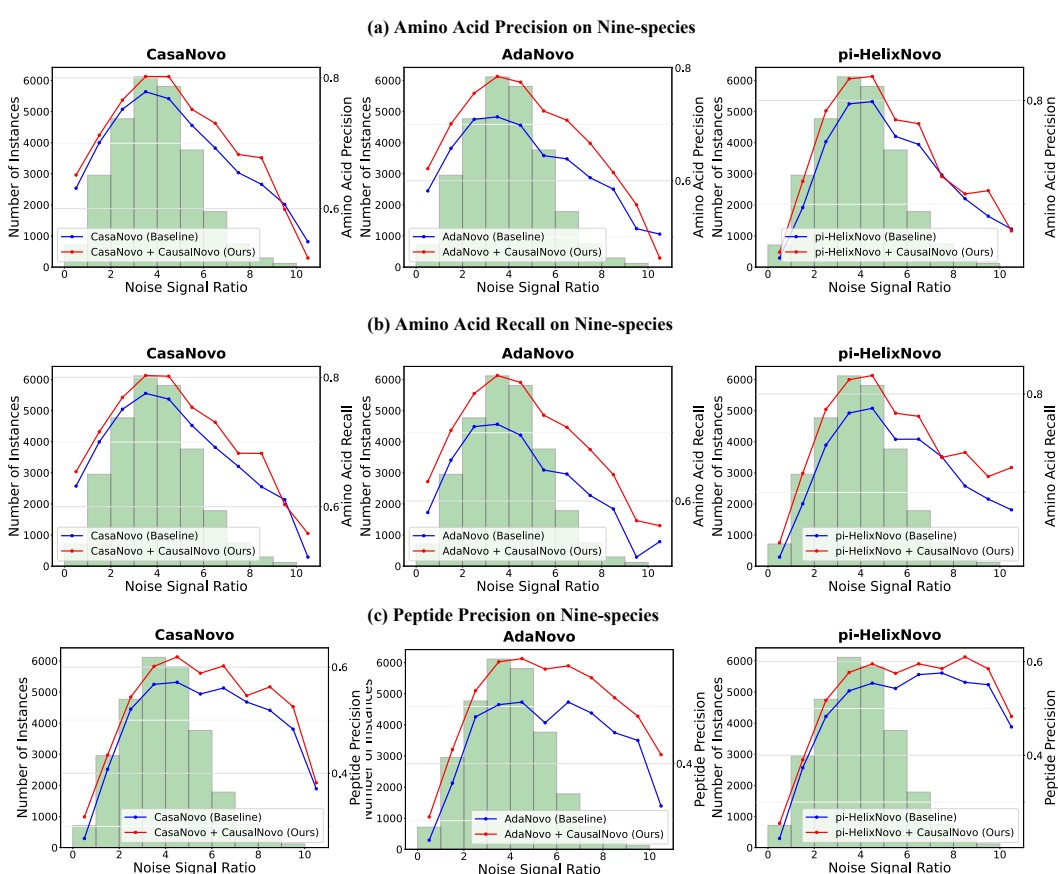

Figure 8: Generalization across varying Noise Signal Ratios (NSR) on the **Nine-species** dataset.

**Robustness evaluation on adding additional noise peaks.** To further evaluate the robustness of the CasualNovo framework against noise peaks, we conduct additional noise addition experiments. Specifically, we artificially introduce 5%, 10%, 20%, and 30% noise peaks (derived from ions outside the 18 theoretical ion types, including singly and doubly charged ions: $b^+$, $y^+$, $a^+$, $b^{2+}$, $y^{2+}$, $a^{2+}$, and neutral loss variants: -NH3 and -H2O for each ion type) to spectra in the test set and compare the baseline with our method. The results in Table 11 show that our framework consistently outperforms the baseline. Furthermore, as the noise increases, the baseline's performance degrades significantly faster, while our method shows greater relative improvement.

**Ablation on the Peak Distinction Strategy.** For the distinction between causal and non-causal peaks, we consider all b, y, and a ions because they are the dominant ions in the benchmark datasets. To test the impact of ion type selection, we also employ a more extensive ion set which contains 18 types of ions (including singly and doubly charged ions: $b^+$, $y^+$, $a^+$, $b^{2+}$, $y^{2+}$, $a^{2+}$, and neutral loss variants: -NH3 and -H2O for each ion type), following Mao et al. (2023). Based on this, we retrain our CausalNovo on the Nine-species dataset. The results in Table 12 show that this method achieves performance comparable to our initial CausalNovo. This is because b, y, and a ions already

Table 9: Comparison with different SSL methods.

| Method | Nine-species | | | HC-PT | | |
|---|---|---|---|---|---|---|
| | AA.Prec | Pep.Prec | Pep.AUC | AA.Prec | Pep.Prec | Pep.AUC |
| CasaNovo (Baseline) | 0.741 | 0.529 | 0.493 | 0.525 | 0.324 | 0.290 |
| + Contrastive Learning | 0.746 | 0.536 | 0.493 | - | - | - |
| + SimSaim | 0.743 | 0.541 | 0.507 | 0.581 | 0.398 | 0.372 |
| + MAE | 0.740 | 0.534 | 0.498 | 0.547 | 0.341 | 0.307 |
| **+ CausalNovo (Ours)** | **0.765** | **0.564** | **0.528** | **0.639** | **0.459** | **0.426** |

Table 10: Experiments on the configuration of the CEM module.

| Configuration | AA.Prec | Pep.Prec | Pep.AUC |
|---|---|---|---|
| CasaNovo (Baseline) | 0.741 | 0.529 | 0.493 |
| Linear Layer | 0.755 | 0.555 | 0.520 |
| 1 Transformer Layer | 0.759 | 0.560 | 0.523 |
| **+ CausalNovo** | **0.765** | **0.564** | **0.528** |

Table 11: Robustness evaluation on adding additional noise peaks to the spectra on the Nine-species and HC-PT datasets. Relative Improvement (RI, %) is calculated as the percentage improvement of our method over CasaNovo for each noise ratio.

| Noise Ratio | Method | Nine-species | | | | HC-PT | | | |
|---|---|---|---|---|---|---|---|---|---|
| | | AA.Prec | RI ↑ | Pep.Prec | RI ↑ | AA.Prec | RI ↑ | Pep.Prec | RI ↑ |
| 0% (Baseline) | CasaNovo | 0.741 | - | 0.529 | - | 0.525 | - | 0.324 | - |
| | + CausalNovo | **0.765** | - | **0.564** | - | **0.635** | - | **0.459** | - |
| 5% | CasaNovo | 0.707 | - | 0.500 | - | 0.506 | - | 0.309 | - |
| | + CausalNovo | **0.733** | **0.41%** | **0.539** | **0.88%** | **0.622** | **1.57%** | **0.447** | **2.02%** |
| 10% | CasaNovo | 0.670 | - | 0.467 | - | 0.479 | - | 0.284 | - |
| | + CausalNovo | **0.701** | **1.22%** | **0.512** | **2.34%** | **0.609** | **4.66%** | **0.436** | **7.32%** |
| 20% | CasaNovo | 0.611 | - | 0.422 | - | 0.446 | - | 0.263 | - |
| | + CausalNovo | **0.650** | **2.51%** | **0.469** | **3.22%** | **0.584** | **7.02%** | **0.414** | **9.03%** |
| 30% | CasaNovo | 0.561 | - | 0.383 | - | 0.421 | - | 0.242 | - |
| | + CausalNovo | **0.608** | **3.77%** | **0.436** | **4.77%** | **0.564** | **8.55%** | **0.400** | **12.48%** |

Table 12: Experiment results on different peak distinction strategies.

| Method | Amino Acid Precision | Peptide Precision | Peptide AUC |
|---|---|---|---|
| CasaNovo (Baseline) | 0.741 | 0.529 | 0.493 |
| + CausalNovo (Ours) | 0.765 | 0.564 | 0.528 |
| + CausalNovo (18 ion types) | 0.764 | 0.561 | 0.526 |

cover most of the peaks in the spectra and provide the most stable signals for peptide identification. Together, these findings confirm that our framework is robust against the distinction strategy.

**More analysis of model attention to causal peaks.** We conduct a systematic analysis of the model's attention to causal versus non-causal peaks during amino acid prediction. Specifically, for each amino acid, we count the number of causal peaks among the top three most attended peaks as determined by the model. Following the methodology of $\pi$-xNovo, we calculate an interpretable attention matrix as $M = \text{Mean}(\text{Concat}(head_i^l \text{ score}))$, where $head_i^l$ score represents the attention score for the $i$-th head in layer $l$. This score is computed as $head_i^l \text{ score} = \text{Attention Score}(D'^l W_i^{lQ}, \hat{S} W_i^{lK})$. Here, $W_i^{lQ}$ and $W_i^{lK}$ denote the learnable weight matrices for the query and key components, $D'^l$ is the intermediate representation in layer $l$ after self-attention, and $\hat{S}$ represents the spectrum data.

Our analysis focuses on cases where the baseline model produces incorrect predictions, but our model successfully corrects them (68,468 amino acids, as summarized in Table 14). In these cases, the baseline model fails to attend to any causal peaks in 14.18% of instances. In contrast, our model reduces this failure rate to 5.44%. Moreover, the proportion of amino acids where all top three

Table 13: Sensitivity to hyper-parameters.

| $\gamma$ | Prec.A | Prec.P | $\alpha$ | Prec.A | Prec.P |
|---|---|---|---|---|---|
| - | 0.741 | 0.529 | - | 0.741 | 0.529 |
| 2 | 0.750 | 0.547 | 0.3 | 0.747 | 0.545 |
| **4** | **0.753** | **0.552** | **0.5** | **0.753** | **0.552** |
| 6 | 0.749 | 0.534 | 0.7 | 0.752 | 0.552 |

attended peaks are causal increases significantly, from 16.36% with the baseline model to 39.85% with our model—an improvement of 23.49%. This demonstrates a substantial enhancement in the model's ability to focus on causal peaks, contributing to its superior prediction accuracy.

Table 14: Systematic evaluation of amino acids where our framework corrects a baseline error.

| # Causal Peaks | 0 | 1 | 2 | 3 |
|---|---|---|---|---|
| Baseline | 9,710 | 23,652 | 23,906 | 11,200 |
| Ratio (%) | 14.18% | 34.54% | 34.92% | 16.36% |
| + CausalNovo | 3,728 | 11,090 | 26,368 | 27,282 |
| Ratio (%) | 5.44% | 16.20% | 38.51% | 39.85% |

**Sensitivity to Hyper-Parameters.** We also evaluated the sensitivity of our CausalNovo framework to the $m/z$ threshold ($\gamma$) and the replacement ratio ($\alpha$). The results are presented in Table 13. The best performance was achieved with $\gamma = 4$ and $\alpha = 0.5$, and the performance remained relatively stable across the tested values. Notably, these experiments were conducted on the Nine-species dataset, and these hyper-parameters are kept consistent across other datasets.

**Qualitative Analysis.** In addition to the systematic evaluation in Table 14, we further conduct a qualitative analysis, as shown in Figure 9. Specifically, we visualize the model's attention corresponding to each peak, thereby quantifying how much each amino acid prediction attends to each spectrum peak. We also highlight the top three most attended peaks and compare the behaviors of the baseline and our model. As shown in the figure, the ground-truth sequence is "VLEPA**V**SAR", while the baseline predicts "VLEPA**S**GLR", making an error at the sixth amino acid. Inspecting the interpretable attention matrix reveals that the baseline's top three attended peaks ($m/z$ values: 129.1021, 867.5272, 110.0711) do not correspond to any causal peaks, which contributes to the misprediction. In contrast, our model's top three attended peaks ($m/z$ values: 678.9623, 217.0837, 207.4436) correspond to the $b_7^+ - NH_3$, $y_4^{2+}$, and $y_4^{2+} - H_2O$ ions, respectively. By focusing on these causal peaks, our model accurately predicts the correct peptide sequence. This shows that our model is capable of shifting the attention toward causal peaks, leading to more accurate predictions.

**Efficiency Analysis.** As shown in Table 15, while CausalNovo introduces negligible inference overhead, it requires approximately 2.3x longer training time than the baseline model due to multiple forward passes per batch. Considering the substantial improvement in performance and robustness, as well as the minimal inference cost in practical applications, this increase in training time is tolerable in practice. However, we acknowledge that this remains a potential limitation, and future work will focus on improving training efficiency.

## A.3 More Details

Finally, we provide the statistics of the three benchmarking datasets in Table 16 to assist in replicating our work. The average peptide length is obtained from NovoBench (Zhou et al., 2024).

## B LLM Usage

Large Language Models (LLMs) were used to aid in polishing of the manuscript. Specifically, we used an LLM to assist in refining the language, improving readability, and ensuring clarity in various sections of the paper. The model helped with tasks such as sentence rephrasing, grammar checking, and enhancing the overall flow of the text.

Table 15: Training and inference time comparison. The training time is measured per epoch with a batch size of 32, while the inference time is evaluated on the test set using a batch size of 256 and a beam search size of 5.

|  | Training Time / Epoch (s) | Inference Time (s) |
|---|---|---|
| Baseline | 3,131 | 15,866 |
| Ours | 7,346 | 15,971 |

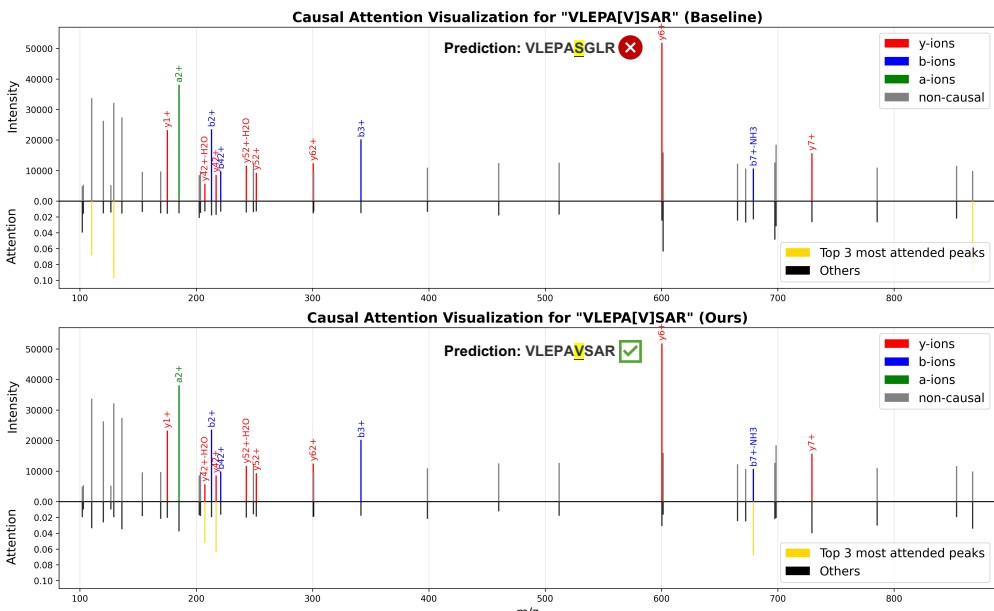

Figure 9: Qualitative comparison when sequencing the spectrum whose ground truth sequence is "VLEPAVSAR". The baseline predicts "VLEPASGLR", which makes a wrong prediction at the sixth amino acid. When examining the interpretable matrix for this prediction, we observe that the baseline's top three attended peaks ($m/z$ values: 129.1021, 867.5272, 110.0711) do not correspond to any causal peaks, leading to the misprediction. In contrast, our model's top three attended peaks are [678.9623, 217.0837, 207.4436], which correspond to the $b_7^+$-$NH_3$, $y_4^{2+}$, $y_4^{2+}$-$H_2O$ ions, respectively. By focusing on these causal peaks, our model successfully predicts the correct peptide sequence "VLEPAVSAR".

It is important to note that the LLM was not involved in the ideation, research methodology, or experimental design. All research concepts, ideas, and analyses were developed and conducted by the authors. The contributions of the LLM were solely focused on improving the linguistic quality of the paper, with no involvement in the scientific content or data analysis.

The authors take full responsibility for the content of the manuscript, including any text generated or polished by the LLM. We have ensured that the LLM-generated text adheres to ethical guidelines and does not contribute to plagiarism or scientific misconduct.

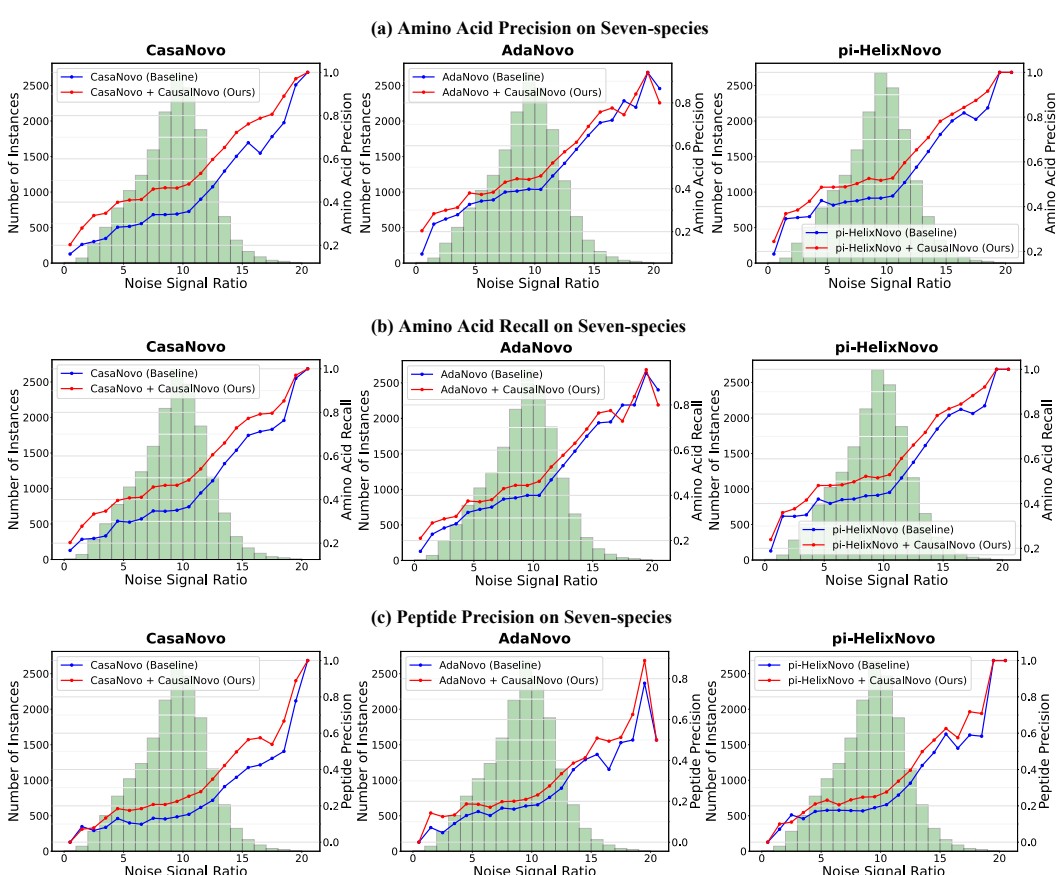

Figure 10: Generalization across varying Noise Signal Ratios (NSR) on the **Seven-species** dataset.

Table 16: The statistics of three public datasets (Zhou et al., 2024).

| Dataset | # Training | # Validation | # Testing | Average Peptide Length | PTM class |
|---|---|---|---|---|---|
| Nine-species (Tran et al., 2017) | 499,402 | 28,572 | 27,142 | 15.01 | 3 |
| Seven-species (Tran et al., 2017) | 317,009 | 17,740 | 17,094 | 15.79 | 3 |
| HC-PT (Eloff et al., 2025) | 213,284 | 25,718 | 26,536 | 12.53 | 1 |

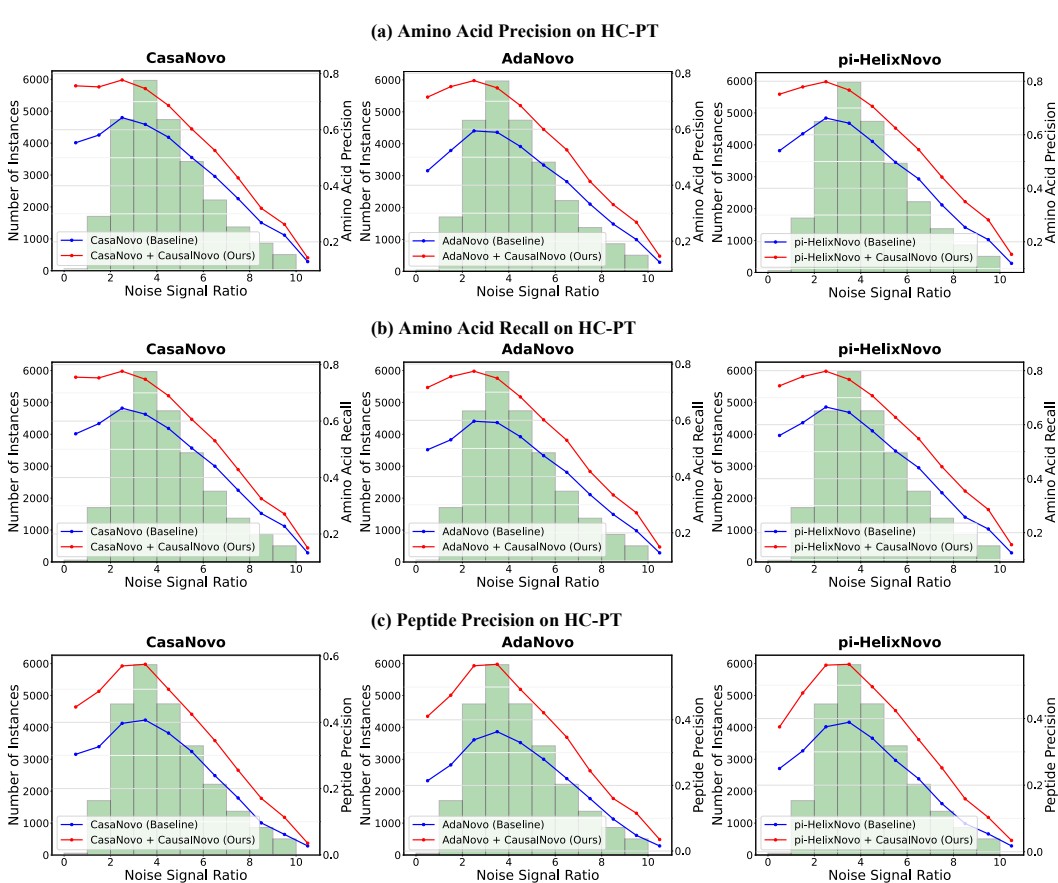

Figure 11: Generalization across varying Noise Signal Ratios (NSR) on the **HC-PT** dataset.

