# OpenReview forum: "CausalNovo: Advancing De Novo Peptide Sequencing via a Causality-Informed Framework"
_ICLR.cc/2026/Conference — ICLR 2026 Conference Withdrawn Submission_

### Official Review · Reviewer_uV6D · 2025-10-17

**Soundness:** 3
**Presentation:** 3
**Contribution:** 2
**Rating:** 6
**Confidence:** 4

**Summary:**

This paper introduces CausalNovo, a model-agnostic framework designed to improve de novo peptide sequencing by learning causal representations from mass spectra.  The authors posit that existing deep learning models often rely on spurious correlations with noise peaks, limiting their robustness and generalizability. CausalNovo addresses this by employing a Structural Causal Model (SCM) to disentangle causal signal information from non-causal noise, guided by the principles of independence and sufficiency.

**Strengths:**

(1) Principled Methodological Framework: The CausalNovo framework is well-grounded in the theory of Structural Causal Models (SCMs) and Reichenbach's Common Cause Principle.

(2) In-depth Model Analysis: The paper goes beyond standard performance metrics by providing insightful analyses that support its central claims.

(3) Clarity and Organization: The manuscript is clearly written and well-structured.

**Weaknesses:**

(1) The framework exhibits notable sensitivity to several key hyperparameters, suggesting that its distinction between “causal” and “non-causal” peaks may depend heavily on parameter choices. This dependence raises concerns about the robustness of the approach and its generalizability to datasets or instruments beyond those tested.

(2) While the authors acknowledge that CausalNovo entails additional computational cost due to multiple forward passes per batch, this overhead is not quantitatively characterized. A clear assessment of the increase in training time or computational resource usage would be essential for evaluating the method’s practicality in real-world scenarios.

(3) The framework’s operational definition of causal ions—limited to theoretical b, y, and a ions—represents a simplified view of fragmentation behavior. In practice, mass spectrometric fragmentation is inherently stochastic, and signals from other ion types or unexpected neutral losses can also carry meaningful information. Although Table 6 addresses this to some extent, the analysis may not fully capture the spectrum of ionization behaviors across different instruments or fragmentation techniques.

**Questions:**

Recently, many transformer-based de novo peptide sequencing methods have emerged [1-4]. The discussion of these approaches in the paper is insufficient.

[1] Bidirectional Representations Augmented Autoregressive Biological Sequence Generation: Application in De Novo Peptide Sequencing.
[2] Universal Biological Sequence Reranking for Improved De Novo Peptide Sequencing.
[3] Latent Imputation before Prediction: A New Computational Paradigm for De Novo Peptide Sequencing.
[4] MassNet: billion-scale mass spectral corpus enables robust de novo peptide sequencing.

---

> ### Author Response · Authors · 2025-11-18
>
> **Weak 1:  About the sensitivity to several key hyperparameters**
>
> We sincerely thank the reviewer for raising this concern. Our framework involves two key hyperparameters: the threshold $\gamma$ for distinguishing causal and non-causal peaks, and the random replacement ratio $\alpha$. To evaluate sensitivity, we conducted experiments on the Nine-species dataset (Appendix Table 13). The results demonstrate that our method is relatively robust to these hyperparameters. For example, reducing $\gamma$ from 4 to 2 only resulted in a 0.3% performance drop, while increasing $\alpha$ from 0.5 to 0.7 led to just a 0.1% decrease.
>
> Kindly note that we did not fine-tune these parameters for each dataset. Instead, we directly applied $\gamma$=4 and $\alpha$=0.5 to the Seven-species and HC-PT datasets. Despite these parameters not being optimized for these datasets, our method still achieved consistent and significant improvements. This highlights the generalizability and robustness of our approach across different datasets and instruments.
>
> **Weak 2:  Suggestion on assessment of training time.**
>
> Thanks for the valuable comment. As shown in **Appendix Table 16**, while CausalNovo introduces negligible inference overhead, it requires approximately 2.3x longer training time than the baseline model due to multiple forward passes per batch. Considering the substantial improvement in performance and robustness, as well as the minimal inference cost in practical applications, this increase in training time is tolerable in practice.  However, we acknowledge that this remains a potential limitation and future work will focus on improving training efficiency.
>
> **Table 1**: Training and inference time comparison. The training time is measured per epoch with a batch size of 32, while the inference time is evaluated on the test set using a batch size of 256 and a beam search size of 5.
>
> | Model    | Training Time / Epoch (s) | Inference Time (s) |
> | -------- | ------------------------- | ------------------ |
> | Baseline | 3,131                     | 15,866             |
> | Ours     | 7,346                     | 15,971             |
>
> **Weak 3: About the definition of causal ions.**
>
> We sincerely appreciate the reviewer's insightful comment. Our causal framework does not restrict the model to using only pre-defined causal ions, but rather guides the model to prioritize the dominant causal signals. In NovoBench datasets, b/y/a ions are primary causal signals, which follow standard fragmentation characteristics in CID/HCD spectra. We also tested an extended 18 ion types definition (Appendix Table 12) and observed nearly identical performance, confirming that our simplified definition captures the primary causal signals effectively. This validates that our framework is robust and that our ion definition is pragmatically sufficient for effective causal modeling.
>
>
> Regarding generalization to different instruments or fragmentation techniques, the fragmentation method is typically known as metadata for the spectra, allowing our framework to naturally adapt the causal ion definition accordingly (e.g., c/z ions for ETD fragmentation). This adaptability allows our approach to be broadly applicable across different mass spectrometry instruments or fragmentation techniques, while maintaining the core principle of causally-informed learning.
>
>
>
> **Question 1: Discussion of recent de novo methods.**
>
> We thank the reviewer for the comment. We have added discussions of these recent methods in the revised Related Work section.

---

### Official Review · Reviewer_aU9q · 2025-10-20

**Soundness:** 3
**Presentation:** 1
**Contribution:** 3
**Rating:** 4
**Confidence:** 3

**Summary:**

This paper proposes a de novo peptide sequence method, named casualNovo, to model the causal relationship between the input mass spectrum and peptide sequence, thereby enabling the model to reduce the interference of noisy data and achieve better sequencing results.

**Strengths:**

1. The method is novel. The application of casualML to denovo is a novel and meaningful application, and the motivation proposed in the article, "these methods are fundamentally limited by their statistical nature，they aim to model dependencies between mass spectra and peptides without accounting for the underlying causal mechanisms." is indeed reasonable.

2. Model-agnostic; CasualNovo is model-agnostic and can be integrated to various denovo models

3. Performance: After applying the model to existing de novo models, the performance improvement is evident, and various experiments have been conducted to fully demonstrate the effectiveness of the model;

**Weaknesses:**

1. The article is hard to follow; the background knowledge introduction about denovo and causal inference is not clear and detailed enough. CasualML is a relatively niche field, and AI for denovo is an even more niche area; readers of this article who are not very familiar with this field will have to spend a lot of time; many details, including "do operation" and "b/y ions", are not explained clearly;
2. There is no significant improvement in performance compared to SOTA models; although the article proves through experiments that applying casualNovo to casanovo and other models can bring improvements, there is no significant difference between the best result (casanovo + casualNovo) and the best baseline (searchNovo), and the performance is worse than the latest SOTA on novobench (ReNovo, LipNovo); it is worth noting that the article omits the latest baseline Lipnovo [1], and mentions ReNovo but does not compare it experimentally. It is unclear why?

[1] Du, Ye, et al. "Latent Imputation before Prediction: A New Computational Paradigm for De Novo Peptide Sequencing." Forty-second International Conference on Machine Learning.

**Questions:**

1. The model is model-agnostic, but it seems that the specific implementation of transferring casualNovo to existing denovo models (such as casanovo, etc.) cannot be determined from the article. Is it by directly adding a new loss? Or is it by using the already trained checkpoint and retraining with casualNovo?

2. Why is there such a significant performance gap between the model performance you retrained (marked with a cross symbol) and the original model performance in Table 1? How can this be explained? Overall, your retrained results are significantly higher than those of the original NovoBench, so the performance improvement may be attributed to your training trick rather than casualNovo.

3. Since peptide precision is the most important metrics, why do Figures 3 and 4 only include amino acid precision?

Overall, the motivation behind the problem statement is meaningful, and the method seems promising. However, certain experimental results and numerous details leave room for confusion and doubt.

---

> ### Author Response · Authors · 2025-11-18
>
> **Weak 1: Concern about the background introduction.**
>
> We sincerely thank the reviewer for the helpful feedback. Due to page limitations, we included the background knowledge in Appendix A.1, which introduces mass spectrum generation (including b and y ions) and the entire de novo sequencing pipeline. We believe this could help readers without a proteomics background better understand the context of our research. Regarding CausalML, although we included a discussion of related work, we appreciate the reviewer’s suggestion to provide more details about the terminology, such as the do-operation. In the revised version, we have added clearer explanations of key CausalML concepts to make the paper more accessible to readers less familiar with this domain.
>
>
> **Weak 2: Concern about the performance compared to SOTA.**
>
> Thank you for the valuable suggestion. The method we propose is a unified framework designed to be integrated into existing models. As a result, our performance comparisons are conducted under the same training settings as the baseline models, rather than focusing solely on presenting a standalone SOTA model.
>
> For LIPNovo, we performed additional experiments by integrating CausalNovo into the method. As shown in Table 1, this integration significantly improves the performance of LIPNovo across datasets. These results not only achieve SOTA performance but also further demonstrate the generalization capability of CausalNovo. Regarding ReNovo, it is fundamentally a retrieval-based method operating during the test time, which means it does not require training on a pre-established model. As such, it represents an orthogonal work to ours. However, our method (built upon LIPNovo) also significantly outperforms ReNovo.
>
> **Table 1**: Integrating our method into LIPNovo.
>
> |         | Peptide      |       |               |       |       |       | Amino acid   |        |               |        |       |        |
> | :------ | ------------ | ----- | ------------- | ----- | ----- | ----- | ------------ | ------ | ------------- | ------ | ----- | :----- |
> |         | Nine-species |       | Seven-species |       | HC-PT |       | Nine-species |        | Seven-species |        | HC-PT |        |
> |         | Prec.        | AUC   | Prec.         | AUC   | Prec. | AUC   | Prec.        | Recall | Prec.         | Recall | Prec. | Recall |
> | ReNovo  | 0.568        | 0.528 | 0.278         | 0.228 | 0.467 | 0.436 | 0.770        | 0.769  | 0.512         | 0.514  | 0.651 | 0.648  |
> | LIPNovo | 0.582        | 0.547 | 0.327         | 0.281 | 0.458 | 0.427 | 0.797        | 0.797  | 0.557         | 0.560  | 0.637 | 0.643  |
> | Ours    | 0.593        | 0.567 | 0.335         | 0.290 | 0.487 | 0.457 | 0.808        | 0.806  | 0.579         | 0.583  | 0.670 | 0.667  |
>
> **Question 1: About the implementation of transferring casualNovo to existing models**
>
> We greatly appreciate the reviewer’s thoughtful question. In practice, we integrate CausalNovo by adding a causal extraction module (between the spectrum encoder and the peptide decoder) and corresponding loss objectives into existing de novo models. The models are then retrained end-to-end from scratch with these components included, rather than fine-tuning from pretrained checkpoints.
>
> **Question 2:  About the performance gap between retrained baseline and original performance**
>
> We appreciate the reviewer’s attention to this detail. In our experiments, we retrained the baseline models to ensure a fair comparison. Importantly, we kept all training and inference configurations consistent, including the learning rate, learning rate schedule, weight decay, batch size, training epochs, beam size, and other hyperparameters related to the spectra. No special training tricks were applied to our model. The higher performance of the retrained baselines compared to the originally reported NovoBench results can likely be attributed to differences in training settings. For example, in the official NovoBench configuration file, the learning rate is set to 5e-5, whereas we used a learning rate of 5e-4. Additionally, differences in hardware or computational devices may have introduced slight variations. We believe that retraining the baselines under consistent conditions ensures a fairer comparison and makes our results more solid and reliable.
>
> **Question 3: Why do Figures 3 and 4 only include amino acid precision ?**
>
> We sincerely appreciate the reviewer’s question. Amino acid precision is presented in Figures 3 and 4 as it provides a more granular and sensitive measure for analyzing the model's behavior, which aligns with the primary focus of these analyses. Due to page limitations, the peptide-level analysis was included in Appendix A.2.

---

> > ### Comment · Reviewer_aU9q · 2025-11-20
> > **reply**
> >
> > Thank you for the author's answer.
> >
> > What I still have doubts about is: Can you ensure that the baseline model (casanovo+cross symbol) of the reproduced version and the casualNovo (casanovo+casualNovo) based on it use the same hyperparameter settings and training techniques? Only in this way can we ensure that the performance improvement comes from casualNovo itself.
> >
> > In addition, the results presented in Figures 3 and 4 do not seem to suggest that the model exhibits better robustness. I would appreciate it if the author could elaborate on how they interpret the analysis results?
> >
> > In the newly added Table 1, the improvement of casualnovo compared to lipnovo is fair (generally 1～2 %).

---

> > > ### Author Response · Authors · 2025-11-20
> > > **Thanks for the feedback**
> > >
> > > We sincerely appreciate the reviewer for the feedback. We address the new comments as follows.
> > >
> > > **Comment 1: Can you ensure that the baseline model (casanovo+cross symbol) of the reproduced version and the casualNovo (casanovo+casualNovo) based on it use the same hyperparameter settings and training techniques?**
> > >
> > > We appreciate this question and apologize for the lack of clarity in our previous response. To ensure a fair comparison, we strictly maintained the same training hyperparameters between the baseline models and our proposed method. Specifically, the hyperparameters used for **CasaNovo** are exactly the same as those for **CasaNovo + CausalNovo**. This consistency was applied to other baselines as well.
> > >
> > > **Comment 2: In addition, the results presented in Figures 3 and 4 do not seem to suggest that the model exhibits better robustness. I would appreciate it if the author could elaborate on how they interpret the analysis results?**
> > >
> > > We appreciate the feedback. In Figure 3, we evaluate the performance of three baseline models and their CausalNovo-enhanced variants under different perturbation levels. A lower perturbation threshold indicates that more identified noise peaks are replaced. We calculate the Relative Improvement (RI), defined as (baseline_drop/baseline)−(CausalNovo_drop/CausalNovo), to evaluate the robustness of CausalNovo. The observation that RI remains positive across all perturbation levels indicates that CausalNovo effectively mitigates performance degradation caused by noise changes, demonstrating greater robustness compared to the baselines. Furthermore, as the perturbation level increases (i.e., the threshold decreases), the RI value shows an upward trend, suggesting that while baseline models suffer significantly due to their reliance on spurious correlations with non-causal ions, CausalNovo remains relatively stable by leveraging causal representations.
> > >
> > > In Figure 4, we evaluate the generalizability of CausalNovo across varying Noise Signal Ratios (NSR), showing that it consistently outperforms the three baseline models across all NSR levels. Notably, CausalNovo yields significant performance gains in the more common settings where the NSR ranges from 2 to 6. This demonstrates that CausalNovo is robust and particularly effective at mitigating the influence of noise ions in typical experimental scenarios.
> > >
> > > **Comment 3: In the newly added Table 1, the improvement of casualnovo compared to lipnovo is fair (generally 1～2 %).**
> > >
> > > We appreciate the feedback. Given that LIPNovo is already a high-performance baseline, our method demonstrates consistent superiority. On the HC-PT dataset in particular, we achieve gains of **3.3%** and **2.4%** in the amino acid precision and recall, underscoring the non-trivial improvement.

---

> > > > ### Author Response · Authors · 2025-11-26
> > > >
> > > > Dear Reviewer aU9q,
> > > >
> > > > We sincerely thank you for your thoughtful insights provided in your review. We deeply appreciate the time and effort you dedicated to improving our work. We would be grateful if you could let us know whether our revisions and responses adequately address your concerns, or if there are any remaining points we can clarify.
> > > >
> > > > Best,
> > > > The authors

---

### Official Review · Reviewer_NVPV · 2025-10-20

**Soundness:** 2
**Presentation:** 3
**Contribution:** 3
**Rating:** 6
**Confidence:** 4

**Summary:**

The paper proposes CausalNovo, a model-agnostic framework for de novo peptide sequencing that injects causal principles (independence & sufficiency) into spectrum-to-peptide models. Concretely, the authors add a Causality Extraction Module (CEM) that masks/weights latent peak representations, perform replace-based interventions on noise peaks inferred via theoretical spectra, and train with contrastive invariance and information-theoretic objectives so that predictions rely on causal fragment ions rather than spurious peaks. Across three public datasets and three strong baselines, the method reports consistent improvements (up to ~10% on AA/peptide/PTM metrics) and reduced vulnerability under noise perturbations and varying noise-signal ratios, with negligible inference overhead (<1%) but extra training cost.

**Strengths:**

- Clear causal motivation & SCM formalization. The paper formulates MS-based sequencing under a simple SCM (variables X, C, S, Y) and derives two practical principles (independence, sufficiency) that directly inform the learning objectives.
- Pragmatic intervention design. Identifying non-causal peaks via proximity to the theoretical spectrum and then replacing a fraction of them with batch-realistic noise is simple, domain-grounded, and avoids distribution shift; adding the theoretical peaks back helps preserve causal links.
- Sound training objectives. Independence is enforced by a symmetric contrastive loss between causal latents before/after intervention; sufficiency/purification are tied to standard cross-entropy on causal/non-causal partitions.
- Robustness analysis. Systematic noise-peak perturbations, NSR sweeps, and attention analysis show reduced reliance on non-causal peaks and better generalization.
- Practicality. The authors have open-sourced code.

**Weaknesses:**

1. Theoretical spectrum definition is oversimplified and not scientifically sufficient.
The method assumes theoretical peaks are only b/y/a ions, but a minimum correct definition in proteomics is six ion types (a/b/c/x/y/z) — and modern SOTA like GraphNovo[1] uses 18+ ion variants. This directly challenges the causal sufficiency assumption claimed in the paper, because relevant causal signal peaks are explicitly omitted.

2. Missing key related works and citations.
GraphNovo[1], ContraNovo[2], RankNovo[3] — all highly relevant and recent — are not cited or discussed.

3. Missing strongest baseline — especially ContraNovo — despite using Nine-species.
If the authors use this benchmark, ContraNovo is a canonical comparison point and must be included. Otherwise, robustness and generality claims remain incomplete.

4. Benchmark choice is unconvincing.
The paper relies mainly on NovoBench, specifically the Nine-species, Seven-species, and HC-PT settings. Taking Nine-species as an example — NovoBench adopts a cross-validation–style train/test split within the same dataset, which is essentially a toy scenario that does not reflect real-world generalization requirements.
In contrast, modern and more realistic evaluation protocols — as adopted by PrimeNovo, ContraNovo, and RankNovo — require training on a large-scale external corpus such as MassiveKB, and evaluating on Nine-species / Seven-species / HC-PT as out-of-distribution test sets.
Under this view, the current evaluation setup does not convincingly demonstrate robustness or generalization. This limitation should be explicitly acknowledged and/or addressed in the discussion section of the paper. The discussion and statement of this limitation in the revised manuscript will be a key factor of my final ratings.

[1] Mitigating the missing-fragmentation problem in de novo peptide sequencing with a two-stage graph-based deep learning model

[2] ContraNovo: a contrastive learning approach to enhance de novo peptide sequencing

[3] Universal Biological Sequence Reranking for Improved De Novo Peptide Sequencing

**Questions:**

These must be addressed to justify acceptance:

1. Why do you assume only b/y/a ions are “causal” — do you believe c/x/z or neutral-loss ions are non-causal?

2. Would your causal extraction still hold after you included the full abcxyz spectrum?

3. Why was ContraNovo not evaluated despite using its canonical benchmark?

4. Can you justify reliance on NovoBench rather than PrimeNovo-style modern setups?

5. Is the causal claim purely architectural, or does it theoretically depend on the correct definition of fragment ion families?

---

> ### Author Response · Authors · 2025-11-18
> **Official Comment by Authors  (Part 1/2)**
>
> **Weak 1:  About theoretical spectrum definition.**
>
> We appreciate the reviewer’s insightful comment. In our implementation, we consider b, y, and a ions because they are the dominant ion types in the benchmark datasets (CID / HCD fragmentation), and are well-established as the primary sequence-informative signals in proteomics literature [1,2].
> We have also tested the sensitivity of our method to ion type selection, employing a more comprehensive set of 18 ion types following GraphNovo (**Appendix Table 12**). The results show that incorporating all 18 ion types yields similar performance, confirming that our simplified definition captures the primary causal signals effectively. This validates that our causal extraction mechanism is robust and that our ion definition is pragmatically sufficient for effective causal modeling.
>
> **Weak 2:  Missing key related works and citations.**
>
> We thank the reviewer for this valuable comment. We have added discussions and citations for GraphNovo, ContraNovo, and RankNovo in the revised Related Work section.
>
> **Weak 3 & Question 3: About the strongest baseline and comparison with ContraNovo.**
>
> We sincerely appreciate the reviewer for this comment. While ContraNovo represents a strong method, it employs a fundamentally different training  data and settings that make direct comparison under our NovoBench setting methodologically problematic. Additionally, we would like to respectively claim that CausalNovo is designed as a model-agnostic framework to enhance existing architectures through causal learning, rather than focusing solely on presenting a SOTA model. To address the concern about comparison with stronger baselines, we further conduct experiment integrating CausalNovo into LIPNovo, the current state-of-the-art under NovoBench settings. As shown in Table 1, CausalNovo further boosts LIPNovo's performance across datasets. These results demonstrate that our framework can provide consistent improvements when integrated with stronger baseline.
>
> **Table 1:** Integrating our method into LIPNovo.
>
> |         | Peptide      |       |               |       |       |       | Amino acid   |        |               |        |       |        |
> | :------ | ------------ | ----- | ------------- | ----- | ----- | ----- | ------------ | ------ | ------------- | ------ | ----- | :----- |
> |         | Nine-species |       | Seven-species |       | HC-PT |       | Nine-species |        | Seven-species |        | HC-PT |        |
> |         | Prec.        | AUC   | Prec.         | AUC   | Prec. | AUC   | Prec.        | Recall | Prec.         | Recall | Prec. | Recall |
> | LIPNovo | 0.582        | 0.547 | 0.327         | 0.281 | 0.458 | 0.427 | 0.797        | 0.797  | 0.557         | 0.560  | 0.637 | 0.643  |
> | Ours    | 0.593        | 0.567 | 0.335         | 0.290 | 0.487 | 0.457 | 0.808        | 0.806  | 0.579         | 0.583  | 0.670 | 0.667  |
>
> **Weak 4 & Question 4: About the benchmark choice.**
>
> We sincerely thank the reviewer for the insightful feedback. We adopted the NovoBench benchmark, which provides a standardized setting that has been widely used to validate the effectiveness of new methods (e.g., AdaNovo, ReNovo, SearchNovo, and LIPNovo). We chose NovoBench because it provides official training datasets with standardized data splits, consistent training configurations, diverse mass spectrometry data, and comprehensive evaluation metrics, making it a fair setting for method evaluation. Notably, the cross-species evaluation setup introduced by DeepNovo ensures no peptide overlap between the training and test sets. This challenging setup serves as an effective benchmark for evaluating the generalization capabilities of an algorithm. As the core of our work is to establish a brand-new methodology, selecting NovoBench was a reasonable choice for our validation and fair comparative analysis.
>
> However, we also agree with the reviewer that the setting adopted by methods such as ContraNovo is more realistic. Training a model on a large-scale, aggregated database like Massive-KB better reflects real-world scenarios and endows the model with greater utility in practical applications.
> We believe that a discussion and statement of this limitation will provide readers with a more balanced perspective on our method and strengthen the paper. Therefore, in accordance with your suggestion, we have added a discussion of this limitation to the Conclusion and Discussion Section of the revised manuscript.

---

> ### Author Response · Authors · 2025-11-18
> **Official Comment by Authors (Part 2/2)**
>
> **Question 1 & 2: About the causal ion definition.**
>
> We appreciate the reviewer’s thoughtful question. We do not assume that c/x/z or neutral‑loss ions are non‑causal. Our choice of b/y/a as primary causal signals follows standard fragmentation characteristics in CID/HCD spectra in NovoBench datasets, which dominate the sequence‑informative signal [1,2].
>
> Our method is robust and remains valid when the full a/b/c/x/y/z ions are included. We employed a more extensive ion set, including 18 types of ions (as in GraphNovo), to verify both the vulnerability experiments (**Table 6**) and robustness analysis (**Appendix Table 12**). These results demonstrate that our causal extraction mechanism still holds under expanded ion definitions, and our consideration is sufficient in practice.
>
>
> **Question 5: Is the causal claim purely architectural, or does it theoretically depend on the correct definition of fragment ion families?**
>
> We thank the reviewer for the insightful question. The causal claim is indeed a synthesis of both architecture and domain knowledge, where one cannot be effective without the other. To be precise, the claim is architectural in *how* it operates but depends on the ion family definitions for *what* it operates on. While the framework is designed to enforce causality, it also requires the definition of fragment ion families to inform the model which specific peaks are causally generated signals that it should focus on. However, it does not require exhaustively including all possible fragmentation ion types. As long as the major ion types that carry sufficient information for peptide sequencing (e.g., b/y/a in CID/HCD spectra) are correctly defined, the causal extraction mechanism remains both theoretically valid and empirically robust.
>
>
>
> **Reference:**
>
> [1] Khatun J, Ramkissoon K, Giddings M C. Fragmentation characteristics of collision-induced dissociation in MALDI TOF/TOF mass spectrometry[J]. Analytical chemistry, 2007, 79(8): 3032-3040.
>
> [2] Lee K W, Peters-Clarke T M, Mertz K L, et al. Infrared photoactivation boosts reporter ion yield in isobaric tagging[J]. Analytical chemistry, 2022, 94(7): 3328-3334.

---

### Official Review · Reviewer_gfnh · 2025-10-29

**Soundness:** 3
**Presentation:** 4
**Contribution:** 3
**Rating:** 8
**Confidence:** 4

**Summary:**

CausalNovo presents a causality-informed framework for de novo peptide sequencing that aims to improve robustness against non-causal spectral noise. By introducing a Causality Extraction Module and applying independence and sufficiency principles through mutual information and contrastive learning objectives, the framework disentangles causal signal peaks from noise in the latent space. The approach consistently improves performance over several Transformer-based baselines, demonstrating better generalization under noisy or perturbed conditions.

**Strengths:**

1. Introduces a novel causality-inspired framework that addresses a real limitation in current de novo peptide sequencing
2. The method integrates causal principles (independence and sufficiency) with practical implementation through contrastive learning, showing consistent performance gains across datasets.
3. The work is well-motivated and provides detailed ablation studies to analyze each component’s contribution.

**Weaknesses:**

1. Although the framework is claimed to yield causal representations, the authors never visualize or interpret what these causal embeddings represent. It would be helpful to include some qualitative examples in addition to the quantitative results in Table 14 to improve interpretability.
2. The authors mention that the method “increases training cost,” but it would be valuable to quantify this overhead and compare it directly with the base models.
3. Does the proposed method effectively work only for Transformer-based variants such as CasaNovo, but not for other architectures like DeepNovo?

**Questions:**

see weakness

---

> ### Author Response · Authors · 2025-11-18
>
> **Weak1: Suggestion to include some qualitative examples to improve interpretability.**
>
> We sincerely appreciate the reviewer for the valuable suggestion. In addition to the systematic evaluation in Table 14,  we further conduct a qualitative analysis, as shown in **Appendix Figure 9**. Specifically, we visualize the model’s attention corresponding to each peak, thereby quantifying how much each amino acid prediction attends to each spectrum peak. We also highlight the top three most attended peaks and compare the behaviors of the baseline and our model. As shown in the figure, the ground-truth sequence is "VLEPA**V**SAR", while the baseline predicts "VLEPA**S**GLR", making an error at the sixth amino acid. Inspecting the interpretable attention matrix reveals that the baseline’s top three attended peaks ($m/z$ values: 129.1021, 867.5272, 110.0711) do not correspond to any causal peaks, which contributes to the misprediction. In contrast, our model’s top three attended peaks ($m/z$ values: 678.9623, 217.0837, 207.4436) correspond to the $b_7^{+}$–$NH_3$, $y_4^{2+}$, and $y_4^{2+}$–$H_2O$ ions, respectively. By focusing on these causal peaks, our model accurately predicts the correct peptide sequence. This shows that our model is capable of shifting the attention toward causal peaks, leading to more accurate predictions.
>
> **Weak 2: It would be valuable to quantify the training cost.**
>
> Thanks for the valuable comment. As shown in **Appendix Table 16**, while CausalNovo introduces negligible inference overhead, it requires approximately 2.3x longer training time than the baseline model due to multiple forward passes per batch. Considering the substantial improvement in performance and robustness, as well as the minimal inference cost in practical applications, this increase in training time is tolerable in practice.  However, we acknowledge that this remains a potential limitation and future work will focus on improving training efficiency.
>
> **Table 1**: Training and inference time comparison. The training time is measured per epoch with a batch size of 32, while the inference time is evaluated on the test set using a batch size of 256 and a beam search size of 5.
>
> | Model    | Training Time / Epoch (s) | Inference Time (s) |
> | -------- | ------------------------- | ------------------ |
> | Baseline | 3,131                     | 15,866             |
> | Ours     | 7,346                     | 15,971             |
>
> **Weak 3: The effectiveness of the proposed model for other architectures like DeepNovo.**
>
> Thank you for the valuable comment. CausalNovo can also be integrated into other architectures such as DeepNovo. Specifically, we incorporate CausalNovo into DeepNovo by inserting the causal extraction module (CEM) between the Spectrum CNN and the LSTM decoder. **Table 2** shows that this integration leads to a substantial performance improvement.
>
> **Table 2**: Integrating our method into DeepNovo.
>
> |          | Amino Acid Precision | Amino Acid Recall | Peptide Precision | Peptide AUC |
> | -------- | -------------------- | ----------------- | ----------------- | ----------- |
> | DeepNovo | 0.492                | 0.433             | 0.204             | 0.136       |
> | Ours     | 0.496                | 0.505             | 0.232             | 0.161       |

---

### Author Response · Authors · 2025-12-01
**Rebuttal Summary**

Dear Area Chair:

We sincerely thank the reviewers for their constructive feedback and the time dedicated to reviewing our work. In our rebuttal, we have addressed the primary concerns with new evidence and experiments, which we summarize below:

**SOTA Integration (LIPNovo)**: We integrated CausalNovo into the state-of-the-art LIPNovo model. This combination outperformed the LIPNovo, proving our framework's effectiveness on SOTA models.

**Model-Agnostic Validation (DeepNovo)**: To demonstrate generality, we applied CausalNovo to the CNN&RNN-based DeepNovo architecture. The results showed substantial performance improvement, confirming that our framework is effective across diverse architectures.

**Robustness of Ion Definition**: We tested a comprehensive set of 18 ion types (referencing GraphNovo) to address concerns about our simplified b/y/a definition. The performance remained consistent, verifying that our method is robust and effectively captures causal signals without requiring exhaustive ion lists.

**Interpretability & Efficiency**: We provided a case study regarding attention visualizations (Appendix Figure 9) showing the model correctly focuses on causal peaks rather than noise. Additionally, we verified that while training cost increases, inference overhead is negligible (<1%).

**Benchmark Justification**: We maintained NovoBench to ensure standardized, fair comparisons between different models but have added a discussion acknowledging the limitations regarding large-scale, real-world training setups as suggested.

We believe these additional results robustly demonstrate the scientific soundness and effectiveness of CausalNovo. We hope this summary assists in your final assessment.

Best regards,

The Authors

---

### Note · Authors · 2026-02-14

I have read and agree with the venue's withdrawal policy on behalf of myself and my co-authors.

---

### Meta-Review · Area_Chair_Kjp1 · 2026-01-06

**Summary:**

The reviewers' concerns primarily centered on the experimental rigor and the attribution of performance gains. A significant critique was the reliance on the NovoBench benchmark, which some reviewers argued is a toy setup compared to modern, large-scale training protocols like MassiveKB, thereby limiting claims of real-world generalization. Skepticism also persisted regarding the source of improvements, with Reviewer aU9q questioning whether gains stemmed from the causal methodology or merely from training tricks and hyperparameter tuning, especially given the marginal improvement over the SOTA baseline (LIPNovo). Furthermore, technical limitations were raised concerning the computational overhead (approximately 2.3x training time) and the dependence on rule-based priors to define causality, which some viewed as essentially domain-specific data augmentation rather than genuine causal discovery.

**Reviewer Concerns:**

Significant concerns remain outstanding regarding the evaluation realism and the magnitude of improvement; specifically, the authors acknowledged but did not rectify the reliance on the smaller NovoBench dataset rather than the industry-standard MassiveKB, leaving the method's utility in large-scale real-world scenarios unproven. Reviewer aU9q remained unconvinced that the marginal performance gains (~1-3% over SOTA with 2.3x training time) definitively validated the causal framework itself rather than being artifacts of implementation or hyperparameter tuning.

**Reviewer Scores:**

Reviewer gfnh and uV6D seem to maintain their scores as they're satisfied by the authors' demonstration. Reviewers NVPV (Score: 6) and aU9q (Score: 4) would likely remain firm or even become more critical, as their reservations align perfectly with the insight that this method is fundamentally contrastive learning wrapped in causal terminology. They would argue that achieving performance gains merely on the smaller NovoBench dataset is insufficient proof of value; without validation on large-scale corpora, it remains unproven whether these causal improvements are genuine generalizable breakthroughs or simply the expected outcome of applying contrastive objectives to a limited, controlled benchmark.

---

### Decision · Program_Chairs · 2026-01-26

Reject